# Translational regulation of protrusion-localized RNAs involves silencing and clustering after transport

Konstadinos Moissoglu[1†], Kyota Yasuda[1,2,3†], Tianhong Wang[1†], George Chrisafis[1], Stavroula Mili[1]*

[1]Laboratory of Cellular and Molecular Biology,Center for Cancer Research, National Cancer Institute, National Institutes of Health, Bethesda, United States; [2]Program of Mathematical and Life Sciences, Graduate School of Integrated Science for Life, Hiroshima University, Higashi-Hiroshima, Japan; [3]Laboratory for Comprehensive Bioimaging, RIKEN Center for Biosystems Dynamics Research, Suita, Japan

**Abstract** Localization of RNAs to various subcellular destinations is a widely used mechanism that regulates a large proportion of transcripts in polarized cells. In many cases, such localized transcripts mediate spatial control of gene expression by being translationally silent while in transit and locally activated at their destination. Here, we investigate the translation of RNAs localized at dynamic cellular protrusions of human and mouse, migrating, mesenchymal cells. In contrast to the model described above, we find that protrusion-localized RNAs are not locally activated solely at protrusions, but can be translated with similar efficiency in both internal and peripheral locations. Interestingly, protrusion-localized RNAs are translated at extending protrusions, they become translationally silenced in retracting protrusions and this silencing is accompanied by coalescence of single RNAs into larger heterogeneous RNA clusters. This work describes a distinct mode of translational regulation of localized RNAs, which we propose is used to regulate protein activities during dynamic cellular responses.

DOI: https://doi.org/10.7554/eLife.44752.001

*For correspondence:
voula.mili@nih.gov

†These authors contributed equally to this work

Competing interests: The authors declare that no competing interests exist.

## Introduction

Targeting of RNA molecules to distinct subcellular destinations has emerged as a widely-used mechanism that controls the cytoplasmic distribution of a large proportion of expressed transcripts in a range of organisms (*Buxbaum et al., 2015*; *Medioni et al., 2012*; *Meignin and Davis, 2010*). Such RNA localization events are functionally important during development, cell-fate decisions, or physiologic responses of somatic cells, including synaptic plasticity and cell migration (*Condeelis and Singer, 2005*; *Holt and Bullock, 2009*; *Holt and Schuman, 2013*; *Yasuda and Mili, 2016*). Localized RNAs are thought to mediate these functional outcomes by directing and restricting protein production in specific subcellular compartments, thus contributing to the molecular compartmentalization of polarized cells. To achieve spatially-controlled protein production, translation of localized RNAs is thought to be suppressed during transport and activated at their final destination (*Besse and Ephrussi, 2008*; *Buxbaum et al., 2015*; *Jung et al., 2014*).

This model is supported by several lines of evidence. Localized RNAs are associated during transport with RNA-binding proteins (RBPs) that function as translational repressors, which block the initiation or elongation steps of translation. For example, the RBPs Khd1 and Puf6 repress the translation of the yeast ASH1 mRNA until it reaches the tip of the emerging yeast bud. Puf6 inhibits translation by binding to the initiation factor eIF5B, thus preventing ribosome subunit binding to the mRNA (*Deng et al., 2008*; *Gu et al., 2004*; *Paquin et al., 2007*). Supporting the coordinate

regulation of RNA transport and translation, certain RBPs perform multiple functions and can operate to both repress translation and support RNA trafficking (*Abaza and Gebauer, 2008*; *Besse and Ephrussi, 2008*). Such multifunctional RBPs include FMRP, a known translational repressor, which can also link its target RNAs to the kinesin motor-based transport machinery (*Davidovic et al., 2007*; *Dictenberg et al., 2008*; *Richter et al., 2015*). Additionally, IMP-1/ZBP-1 is required both for transport as well as for translational repression of the beta-actin mRNA, by blocking recruitment of the 60S ribosomal subunit and initiation of translation (*Condeelis and Singer, 2005*; *Hüttelmaier et al., 2005*).

Once the RNA reaches its final destination, spatial control of gene expression is achieved by local de-repression of translation. Exemplifying this notion, at the yeast bud tip, the Puf6 and Kdh1 RBPs are phosphorylated by the Yck1 kinase, leading to release of eIF5B and translation activation (*Deng et al., 2008*). Similarly, phosphorylation of ZBP1 by the Src kinase disrupts ZBP1-RNA binding leading to translation of beta-actin RNA (*Hüttelmaier et al., 2005*).

Various other mechanisms maintain localized RNAs in a silenced state, and are relieved in a spatial manner. The Drosophila *nanos* mRNA is deadenylated and translationally repressed in the bulk cytoplasm of Drosophila embryos through the action of the RBP Smaug and the CCR4/NOT deadenylase. At the posterior pole, the Oskar protein relieves this inhibition and leads to de-repression of *nanos* translation (*Jeske et al., 2011*; *Zaessinger et al., 2006*). In neuronal dendrites, translation of RNAs can be suppressed by miRNAs (*Schratt et al., 2006*), and degradation of components of the RISC complex controls synaptic protein synthesis (*Ashraf et al., 2006*).

Transported RNAs can also be maintained in a translationally-repressed state through oligomerization or multiplexing into higher-order RNP particles or granules (*Carson et al., 2008*; *Chekulaeva et al., 2006*; *De Graeve and Besse, 2018*). These particles (also referred to, in the case of neurons, as neuronal transport granules) share protein components as well as liquid-droplet properties with other phase-separated RNA granules, such as P-bodies and stress granules (*De Graeve and Besse, 2018*; *Gopal et al., 2017*). Containment within such granules is thought to retain RNAs in a repressed state, inaccessible to the translation machinery. Local signals can release such 'masked' RNAs and allow their translation (*Buxbaum et al., 2014*; *Kotani et al., 2013*).

We have been investigating a group of RNAs that are localized at protrusions of migrating cells. We refer to these RNAs as 'APC-dependent' because their localization requires the tumor-suppressor protein APC (*Mili et al., 2008*; *Wang et al., 2017*). Localization of APC-dependent RNAs at protrusions requires a particular subset of modified microtubules, namely detyrosinated microtubules, and is mechanically regulated in response to the stiffness of the extracellular environment (*Wang et al., 2017*; *Yasuda et al., 2017*). Specifically, increased actomyosin contractility on stiff substrates, through activation of a signaling pathway involving the RhoA GTPase and its effector formin mDia, leads to formation of a detyrosinated microtubule network, which in turn supports RNA localization at protrusions. Localization of APC-dependent RNAs at protrusions is important for efficient cell migration (*Wang et al., 2017*). We hypothesize that the positive effect of APC-dependent RNAs on cell migration is mediated through local RNA translation at protrusions.

Here, we use polysome association, single-molecule translation imaging reporters, and in situ imaging of endogenous nascent proteins to determine whether APC-dependent RNAs are translated at protrusions and whether their translation is affected by their location in the cytoplasm. We find that indeed, localized RNAs are translated at protrusions, but interestingly they are also translated with similar efficiency regardless of their location within the cell. Intriguingly, we observe that continuous transport to the periphery leads to coalescence of single RNAs into larger clusters that are translationally silenced. We further show that such silencing and clustering occurs at retracting protrusions. Therefore, in contrast to the model described above, APC-dependent RNAs are not locally activated solely at protrusions. Instead, after transport to the periphery, and upon protrusion retraction, they become translationally silent and segregate into multimeric RNA granules. We propose that this mechanism is used to regulate protein activities during dynamic cellular responses.

# Results

## Disrupting the localization of APC-dependent RNAs at protrusions does not affect their translation

As a first step towards assessing whether localization of APC-dependent RNAs at protrusions is coupled to their translation status, we disrupted RNA localization at protrusions and determined whether that affected the efficiency of their translation. To measure translation efficiency, we fractionated cell extracts on sucrose gradients to resolve RNAs according to the number of bound ribosomes (*Figure 1A*). To facilitate a larger scale analysis, we divided each gradient into four fractions based on UV absorbance traces. Fraction one includes free RNPs and the 40S and 60S ribosomal subunits, fraction 2 includes 80S monosomes, and fractions 3 and 4 include light and heavy polysomes, respectively. mRNAs in fractions 1 and 2 largely correspond to non-translated mRNAs,

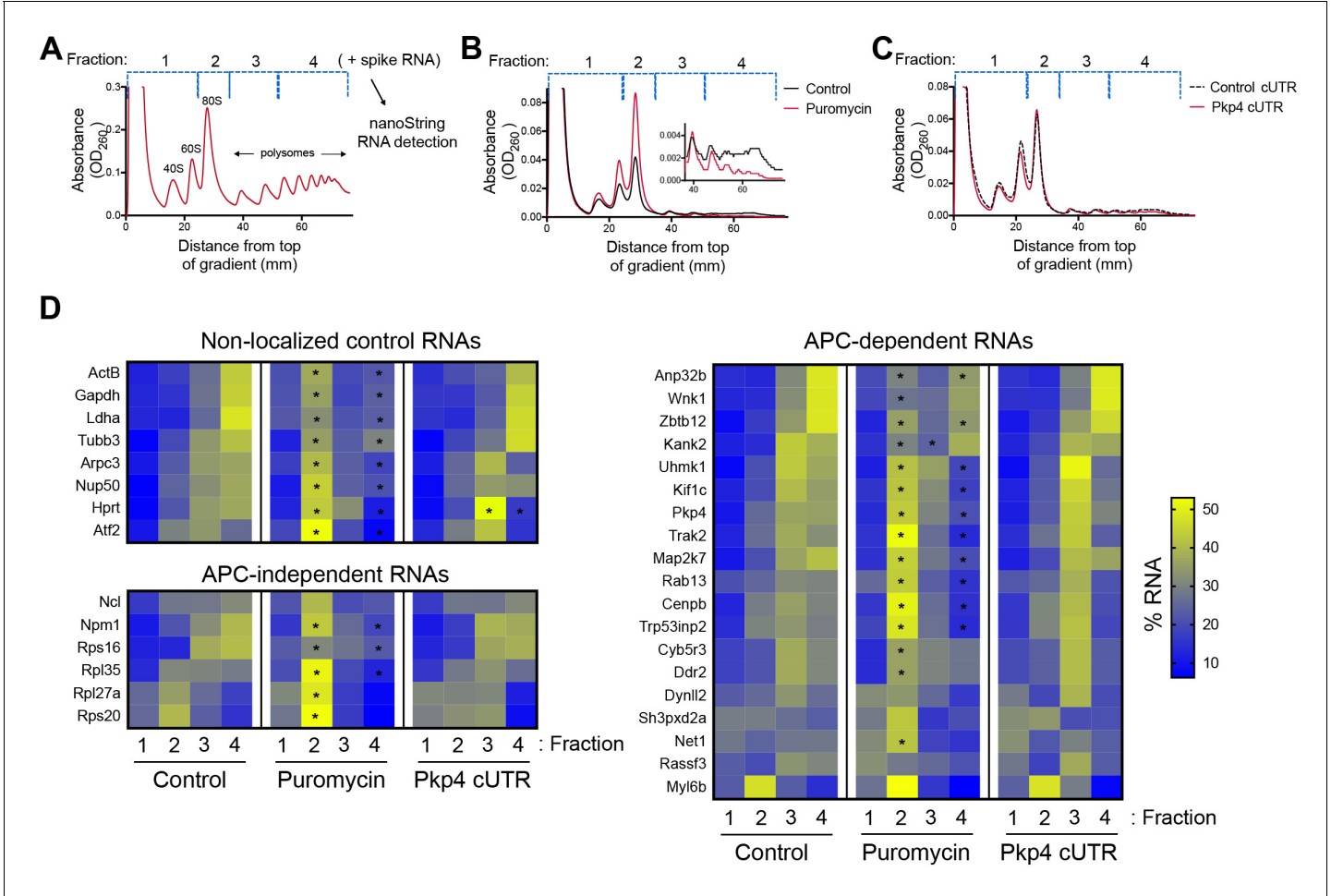

**Figure 1.** Disrupting localization of APC-dependent RNAs, through competition, does not alter their translation. (**A**) Outline of experimental procedure. Sucrose gradients are divided into four fractions based on UV absorbance, an equal amount of spike RNA is added to each, and RNA presence is quantitatively assessed with nanoString analysis. (**B, C**) Representative absorbance profiles of polysome gradients of control, puromycin treated (**B**) or Pkp4-cUTR-expressing cells (**C**). Inset in (**B**) shows an enlargement of the polysome region. (**D**) Heat maps showing RNA presence in polysome gradient fractions, based on nanoString analysis, under the indicated conditions. Gene names are shown on the left. Values indicate averages of 3 independent experiments. Statistically significant differences compared to the corresponding control fractions are indicated by asterisks (2-way ANOVA with Dunnett's multiple comparisons test).

DOI: https://doi.org/10.7554/eLife.44752.002

The following source data is available for figure 1:

**Source data 1.** File containing values used for generation of the heatmaps and statistics of *Figure 1D*.

DOI: https://doi.org/10.7554/eLife.44752.003

whereas mRNAs in fractions 3 and 4 are actively translated. To correct for variations introduced by sample manipulation during RNA purification, an equal amount of in vitro transcribed spike RNA was added to each fraction. The recovered RNA from each fraction was then used to simultaneously detect the levels of multiple RNA species along the four fractions (*Figure 1A*). For detection, we used nanostring analysis which allows for direct RNA counting and thus avoids biases introduced by reverse transcription and amplification. We designed probes to detect 20 protrusion-enriched RNAs, that we have previously defined as APC-dependent, 6 RNAs encoding ribosomal proteins or ribosome biogenesis factors, which we have previously shown are also enriched at protrusions but in an APC-independent manner, and eight control RNAs which based on our prior analysis are not enriched at protrusions (*Wang et al., 2017*) (*Figure 1D*).

To first validate whether our experimental approach can detect changes in translation, we isolated polysome gradient fractions from control cells or cells treated with puromycin, an antibiotic that terminates translation and dissociates polysomes. Indeed, puromycin treatment led to an increase in monosomes and ribosomal subunits and a decrease in heavy polysomes (*Figure 1B*). (Note that NIH/3T3 cells used in these experiments have a reduced baseline amount of ribosomes engaged in translation (i.e. in light and heavy polysomes) compared to HEK293 cells shown in *Figure 1A*). Nanostring analysis of recovered RNA revealed that under control conditions a higher proportion of most RNAs exists in fractions 3 and 4, while treatment with puromycin significantly shifted the distribution of RNAs, in all three groups examined, towards fractions 1 and 2 (*Figure 1D* and *Figure 1—source data 1*). These results therefore indicate that indeed mRNAs present in fractions 3 and 4 correspond to actively translated transcripts, and furthermore, that our nanostring-based methodology can quantitatively detect changes in translation efficiency.

To disrupt the localization of APC-dependent RNAs at protrusions, we employed two different methodologies. One is a competition-based method that relies on overexpression of an exogenous construct carrying the 3′UTR of the APC-dependent RNA, Pkp4. Our prior characterization showed that inducible expression of such competitive-UTR (cUTR) constructs leads to a preferential mislocalization of APC-dependent RNAs from protrusions (see *Wang et al., 2017* for a detailed characterization of the stable cell lines used). A second method relies on the requirement for detyrosinated microtubules for localization of APC-dependent RNAs at protrusions (*Wang et al., 2017*; *Yasuda et al., 2017*). Brief treatment with the chemical compound parthenolide disrupts detyrosinated microtubules (*Yasuda et al., 2017*) and significantly reduces RNA localization at protrusions, measured through the degree of RNA enrichment in isolated protrusion samples (*Figure 2A*).

To test whether mislocalization from protrusions is accompanied by changes in translation, polysome gradient fractions were isolated from control and cUTR- or parthenolide-treated cells and analyzed for the presence of various RNA species. In both cases, cUTR overexpression or parthenolide treatment did not affect the overall gradient profile, suggesting that no overt changes in translation resulted from either treatment (*Figures 1C* and *2B*). Furthermore, both approaches did not result in significant changes in the translation state of the control RNAs, the APC-independent RNAs or the mislocalized APC-dependent RNAs analyzed (*Figures 1D* and *2B* and *Figure 1—source data 1*, *Figure 2—source data 1*). Therefore, disrupting the peripheral localization of APC-dependent RNAs does not affect their translational efficiency, suggesting a lack of coupling between RNA transport and translational control.

We note that both cUTR overexpression and parthenolide treatment resulted in a small apparent reduction in the amount of APC-dependent RNAs sedimenting in the heavy polysome fraction (fraction 4) and an apparently corresponding increase of RNA amounts in polysome fraction 3. These changes are small, they are not statistically significant and do not alter the conclusion that APC-dependent RNAs remain translationally active. Nevertheless, we make a note of this observation because it is manifested quite consistently by APC-dependent RNAs under conditions that disrupt their peripheral localization. In light of data presented below, we believe that this might reflect a real change in the organization of a small fraction of these RNPs (see discussion).

## Single-molecule translation reporters of protrusion-localized RNAs

While the above results suggest that transport and translation of APC-dependent RNAs are not coordinated, they also raise the possibility that assessing the translation status in a whole-cell extract derived from heterogeneous cell populations might not offer the required sensitivity to detect local changes occurring on a single-cell level. To address this, we took advantage of the recently

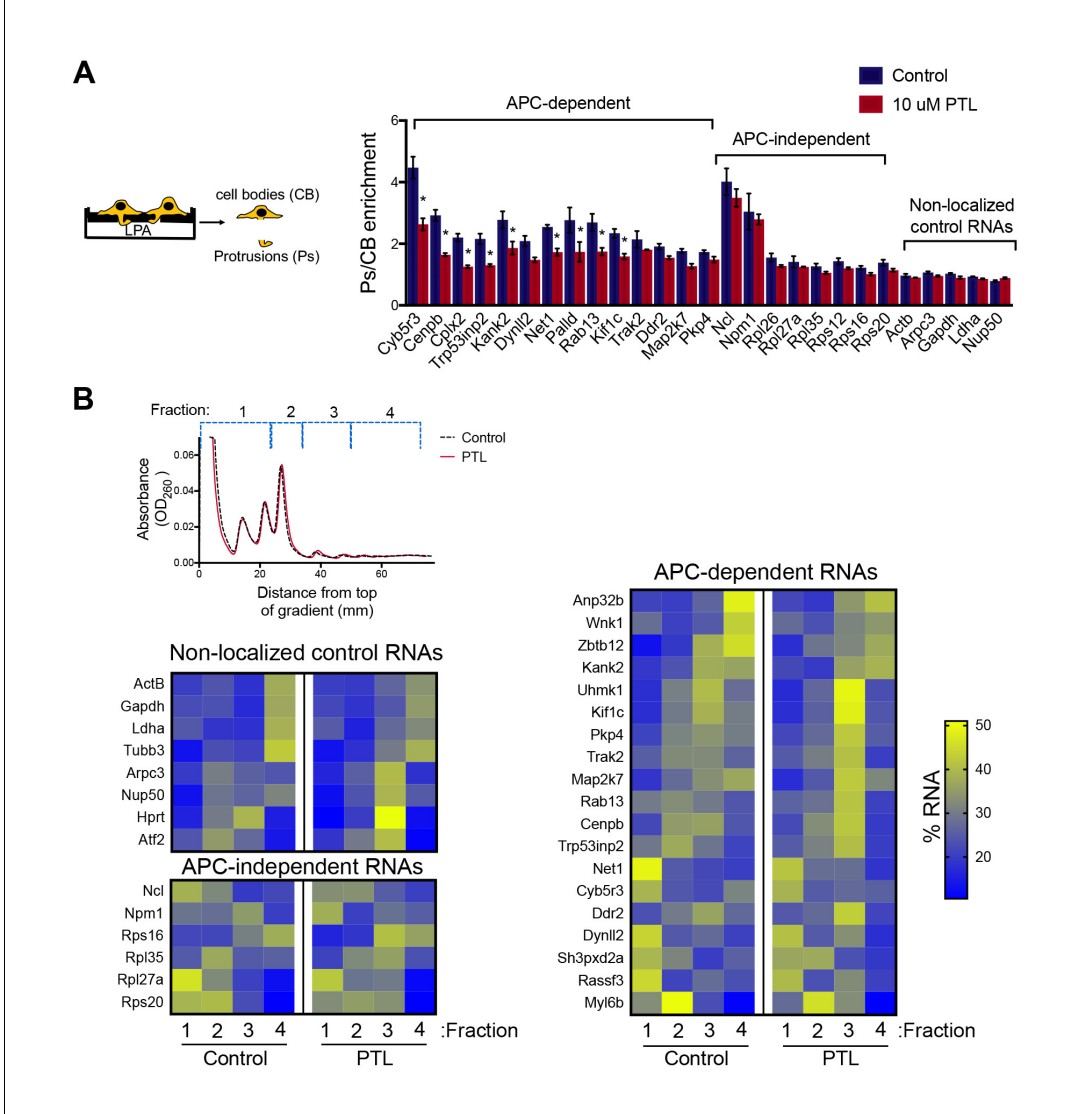

**Figure 2.** Disrupting localization of APC-dependent RNAs, through perturbation of detyrosinated microtubules, does not alter their translation. (**A**) Schematic on the left indicates experimental procedure used for isolation of protrusions. Migration of cells through microporous filters was induced by addition of LPA and protrusion (Ps) and cell body (CB) samples were isolated from control or parthenolide (PTL) treated cells. The indicated RNAs were detected through nanoString analysis to calculate Ps/CB enrichment ratios (n = 3; error bars: standard error). *: p-value<0.04 by two way ANOVA with Bonferroni's multiple comparisons test against the corresponding control. Parthenolide treatment specifically reduces the enrichment of APC-dependent RNAs at protrusions. (**B**) Representative absorbance profiles of polysome gradients of control and PTL-treated cells, and heat maps showing RNA presence in polysome gradient fractions, based on nanoString analysis. Gene names are shown on the left. Values indicate averages of 3 independent experiments. No statistically significant differences were detected by 2-way ANOVA with Dunnett's multiple comparisons test against the corresponding control fractions.

DOI: https://doi.org/10.7554/eLife.44752.004

The following source data is available for figure 2:

**Source data 1.** File containing values used for generation of the heatmaps and statistics of *Figure 2B*.
DOI: https://doi.org/10.7554/eLife.44752.005

developed SunTag-based reporters that allow imaging of translation of single RNA molecules in live cells (*Morisaki et al., 2016*; *Wang et al., 2016*; *Wu et al., 2016*; *Yan et al., 2016*) (*Figure 3A*). These reporters carry a coding sequence which includes, at the 5' end, a series of SunTag peptide epitopes, which are recognized by a single chain antibody fragment fused to superfolder-GFP (scFv-GFP). The concentration of GFP-fused antibodies on a series of nascent peptides generated during

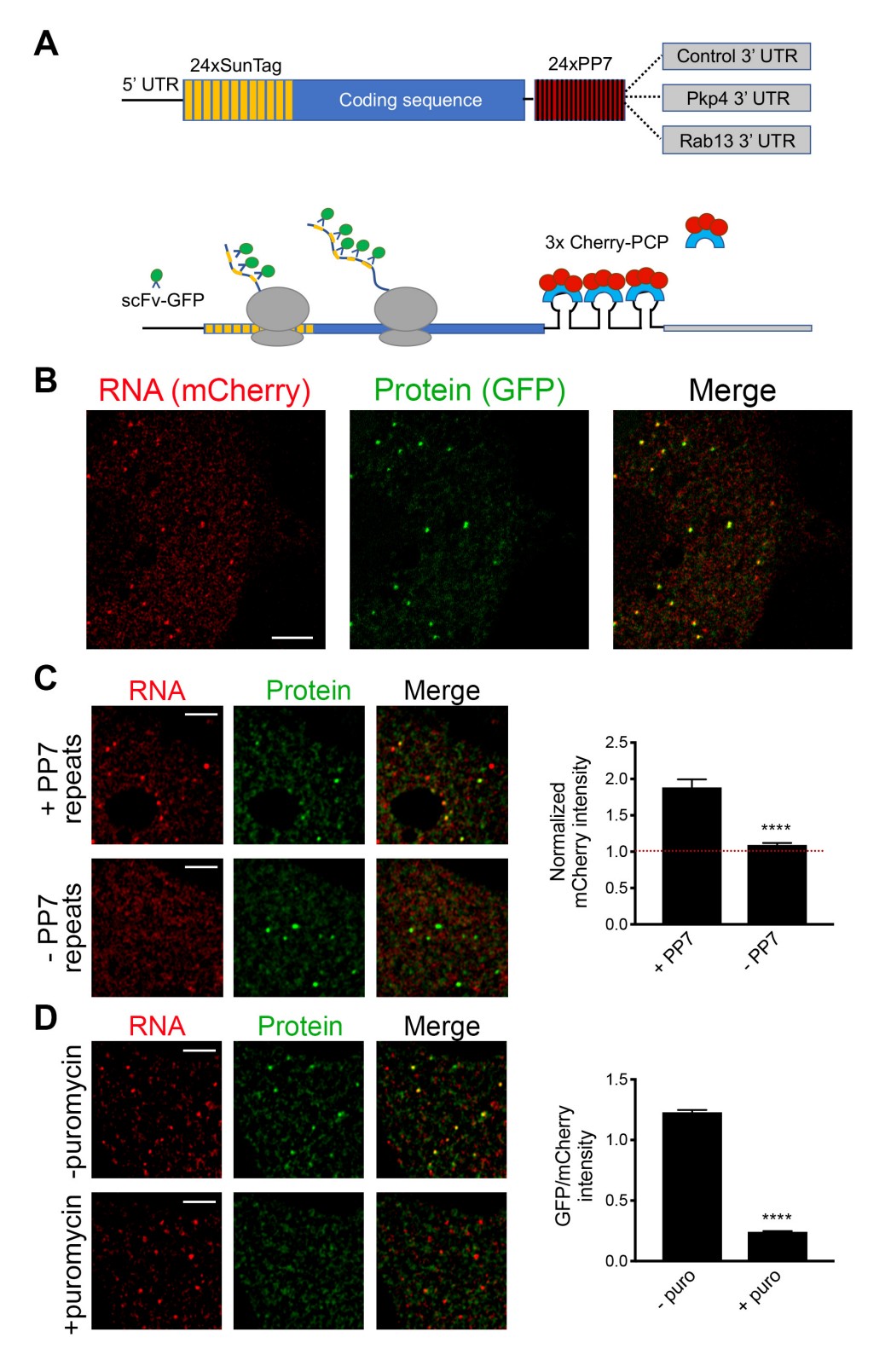

**Figure 3.** Validation of single-molecule translation reporter assay. (A) Schematic of translation reporter constructs for labeling of RNA and nascent protein chains. (B) Live cell imaging snapshot of a cell expressing the control translation reporter. The mCherry channel detects the 3x-mCherry-PCP protein. Bright spots correspond to RNA molecules. Diffuse signal results from free 3x-mCherry-PCP. The GFP channel detects the scFv-GFP antibody. Bright spots overlap with RNA spots (merge image) and correspond to nascent protein at translation sites. Diffuse signal results from free scFv-GFP or

*Figure 3 continued on next page*

*Figure 3 continued*

scFv-GFP bound to the reporter protein released after translation. (C) Cells expressing the control translation reporter, containing PP7 repeats (+PP7), or a reporter without PP7 repeats (-PP7), were imaged live. mCherry intensity overlapping with translation sites (GFP spots) was measured and normalized to the intensity observed in nearby cytoplasmic regions with diffuse signal. Value of 1 indicates that there is no mCherry concentration at translation sites. (D) The same cytoplasmic areas, of cells expressing the control translation reporter, were imaged before and after puromycin addition. GFP/mCherry intensity of individual spots was calculated as a measure of translational efficiency. n > 100 (C) and n > 500 (D) spots from multiple cells observed in three independent experiments; error bars: standard error; ****: p-value<0.0001 by Student's t-test. Scale bars: 5 μm.
DOI: https://doi.org/10.7554/eLife.44752.006

The following figure supplement is available for figure 3:

**Figure supplement 1.** Translation signal of localized reporters reflects active translation.
DOI: https://doi.org/10.7554/eLife.44752.007

translation allows the observation of translation sites as bright GFP spots, which can be distinguished from free antibody molecules or mature proteins released after translation. A second element incorporated in these reporter constructs is a series of binding sites for the PP7 bacteriophage coat protein introduced after the end of the coding sequence. These hairpin elements bind to a PP7 coat protein fused to 3 copies of mCherry fluorescent protein (3x-mCherry-PCP), thus allowing visualization of RNA molecules as bright red spots (*Yan et al., 2016*) (*Figure 3B* and *Video 1*). To use this system, we have generated mouse NIH/3T3 fibroblast cell lines that stably express the GFP-fused antibody and the 3x-mCherry-PCP, and additionally can be induced to express reporter RNAs after addition of doxycycline (*Figure 3A,B* and see below). For our studies, we have been using a control reporter RNA which carries, after the PP7 repeats, a UTR sequence that doesn't direct transport to protrusions, and two localized reporter RNAs in which the PP7 repeats are followed by the 3'UTR of either the mouse Rab13 or Pkp4 RNAs, which as we have previously shown are sufficient to target reporter RNAs to protrusions (*Figure 3A*) (*Mili et al., 2008*; *Wang et al., 2017*).

We first validated the RNA and translation signals detected with our implementation of the assay. To assess the specificity of the detected RNA spots, we compared the signal observed upon expression of the control PP7-containing reporter RNA to that of a similar reporter carrying a deletion of the PP7-binding sites (−PP7) (*Figure 3C*). mCherry intensity overlapping with translation sites (GFP spots) was measured and normalized to the intensity observed in nearby cytoplasmic regions with diffuse signal. The −PP7 construct exhibited a normalized mCherry intensity around 1, indicating that signal intensity overlapping with translation spots was similar to that of the surrounding cytoplasm. By contrast, the +PP7 containing reporter exhibited a significantly higher intensity, indicating that the mCherry signal overlapping translation sites was due to specific recognition of RNA molecules by the fused PP7 coat protein (*Figure 3C*). To assess whether GFP spots are indeed reflecting translation sites, images were acquired before and after treatment with puromycin (*Figure 3D*). RNA spots were identified in the images, and mCherry and GFP intensity in the corresponding regions was recorded. The normalized GFP intensity was calculated as a measure of the translational efficiency of each RNA spot. Puromycin treatment significantly reduced the GFP intensity of RNA spots, confirming that it reflects the presence of nascent protein chains (*Figure 3D*). To further determine whether this concentration of nascent

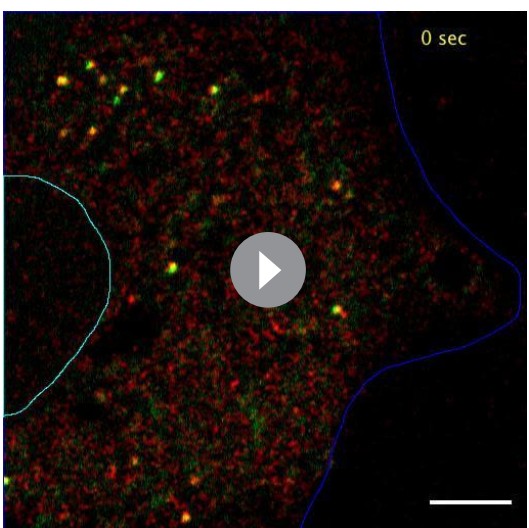

**Video 1.** NIH/3T3 cell expressing scFv-GFP (green), 3x-mCherry-PCP (red) and the control translation reporter. Frames were acquired sequentially and with no time delay, for the duration of the movie (45 s). A merged image of the two channels is shown. Overlapping red and green spots indicate translation sites. Blue line: cell outline. Cyan line: nucleus outline. Scale bar: 5 μm. Single frames of this movie are shown in *Figure 3B*.
DOI: https://doi.org/10.7554/eLife.44752.008

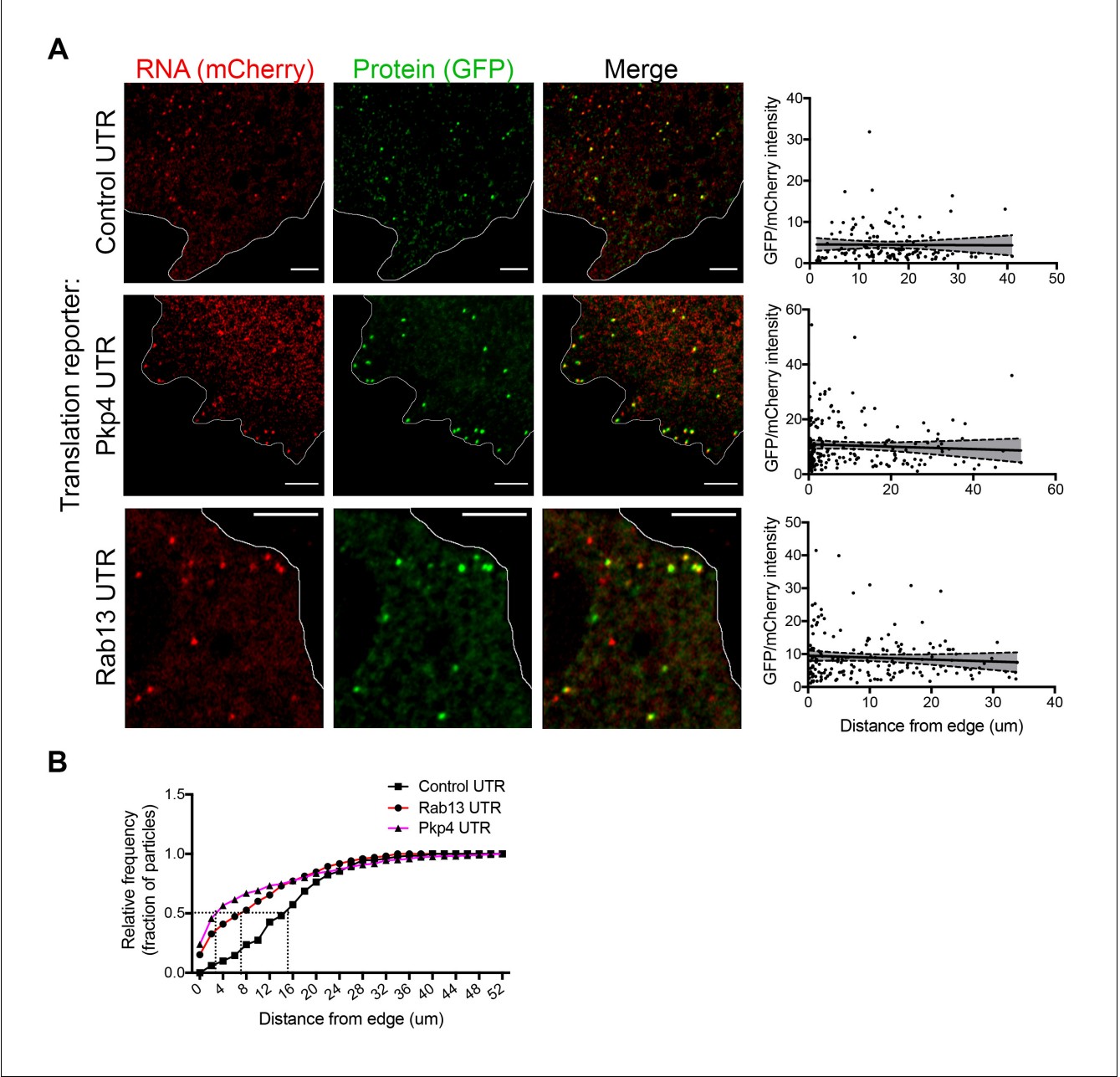

**Figure 4.** RNAs targeted to protrusions are similarly translated in both internal and peripheral locations. (A) Live imaging snapshots of cells expressing the indicated translation reporters. GFP/mCherry intensity of individual spots (indicating translation efficiency) was plotted as a function of distance from the cell edge. More than 200 particles were analyzed from approximately 20 cells. Best fit curves with 95% confidence intervals are overlaid on the graphs. Scale bars: 5 µm. (B) Cumulative frequency distribution plot of translation reporter particles (from panel A) with increasing distance from the cell edge.

DOI: https://doi.org/10.7554/eLife.44752.009

The following figure supplements are available for figure 4:

**Figure supplement 1.** Expression levels of translation reporters and comparison with live-cell imaging.

DOI: https://doi.org/10.7554/eLife.44752.010

**Figure supplement 2.** Intensity histograms of translation reporter particles.

DOI: https://doi.org/10.7554/eLife.44752.011

**Figure supplement 3.** Examples of directionally persistent particles.

DOI: https://doi.org/10.7554/eLife.44752.012

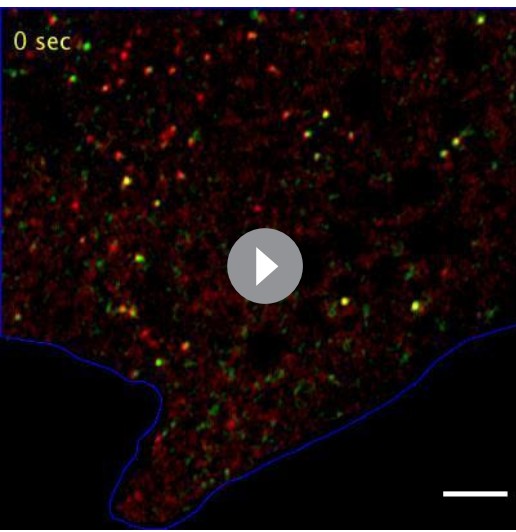

**Video 2.** NIH/3T3 cell expressing scFv-GFP (green), 3x-mCherry-PCP (red) and the control translation reporter. Frames were acquired sequentially and with no time delay, for the duration of the movie (13 s). A merged image of the two channels is shown. Blue line: cell outline. Scale bar: 5 μm. Single frames of this movie are shown in *Figure 4A* (upper panels).
DOI: https://doi.org/10.7554/eLife.44752.013

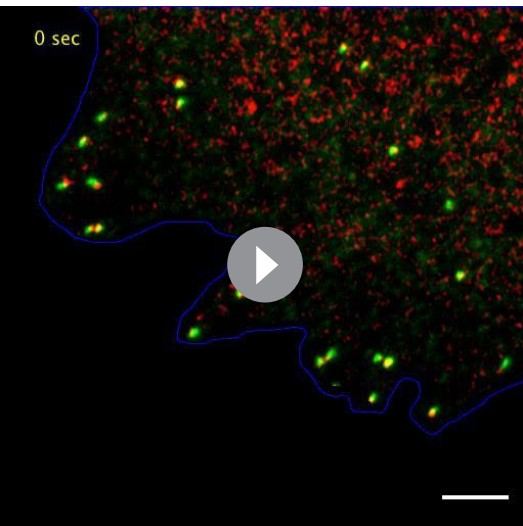

**Video 3.** NIH/3T3 cell expressing scFv-GFP (green), 3x-mCherry-PCP (red) and the localized translation reporter carrying the Pkp4 UTR. Frames were acquired sequentially and with no time delay, for the duration of the movie (36 s). A merged image of the two channels is shown. Blue line: cell outline. Scale bar: 5 μm. Single frames of this movie are shown in *Figure 4A* (middle panels). RNAs are translated both in the periphery and internal regions.
DOI: https://doi.org/10.7554/eLife.44752.014

chains reflects active translation, and not stalled ribosomes, we acquired images of localized translation reporters before and after treatment with harringtonine or lactimidomycin, which block initiating ribosomes but allow elongating ribosomes to run off (*Ingolia et al., 2011*; *Lee et al., 2012*). As an additional control we treated cells with cycloheximide, which stalls elongating ribosomes and prevents release of nascent chains. Indeed, brief 15 min treatment with harringtonine or lactimidomycin, but not cycloheximide, significantly reduced the observed translation signal indicating that it reflects the presence of actively translating ribosomes (*Figure 3—figure supplement 1*).

## RNAs targeted to protrusions are similarly translated in both internal and peripheral locations

We further imaged cells expressing either control or localized reporters (carrying the Rab13 or Pkp4 UTRs) (*Figure 4A* and *Videos 2–4*). Imaging was performed ca. 2 hr after doxycycline induction to ensure that reporter RNAs do not accumulate to high levels and do not deplete the cytoplasmic pools of scFv-GFP antibody and 3x-mCherry-PCP (*Figure 4—figure supplement 1A*). Consistent with the ability of the Rab13 and Pkp4 UTRs to target RNAs to protrusions (*Mili et al., 2008*; *Wang et al., 2017*), a higher proportion of localized reporter RNAs (containing the Rab13 or Pkp4 UTRs) were observed closer to the periphery compared to the control reporter (*Figure 4B*). Around 50% of observed localized reporter molecules were found within 3–7 μm of the cell edge, compared to 15 μm for the control reporter (*Figure 4B*). For all reporters, the majority of RNA particles exhibit mCherry intensities centered around a single peak indicating that they largely exist as single molecules (*Figure 4—figure supplement 2*) (but see also below). The number of particles detected by live-cell imaging is lower than the number detected in fixed cells by FISH (*Figure 4—figure supplement 1B*). The additional RNAs detected by FISH likely correspond to fast-moving molecules that cannot be discerned during live imaging with our current acquisition speed. Observing RNAs during short time-lapse imaging (~20–30 s) reveals that the majority of RNAs are static or exhibit an oscillatory type of motion (*Videos 2–4*). A small subset exhibits short directed movements that might be indicative of active transport (*Videos 5* and *6*; and see below). These observations are consistent with the motion characteristics described for other localized transcripts

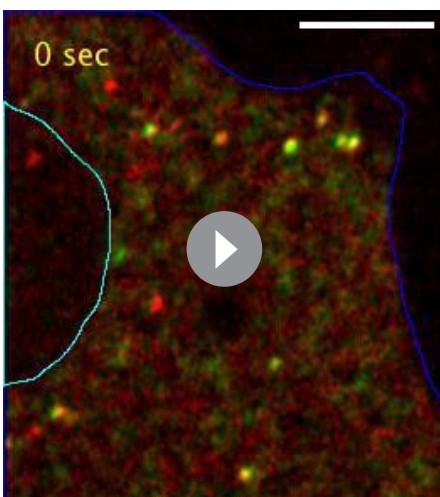

**Video 4.** NIH/3T3 cell expressing scFv-GFP (green), 3x-mCherry-PCP (red) and the localized translation reporter carrying the Rab13 UTR. Frames were acquired sequentially and with no time delay, for the duration of the movie (63 s). A merged image of the two channels is shown. Blue line: cell outline. Cyan line: nucleus outline. Scale bar: 5 μm. Single frames of this movie are shown in *Figure 4A* (bottom panels). RNAs are translated both in the periphery and internal regions.

DOI: https://doi.org/10.7554/eLife.44752.015

(*Dynes and Steward, 2007*; *Gopal et al., 2017*; *Park et al., 2014*; *Yoon et al., 2016*). Given the limitations in imaging and discerning fast-moving molecules, we have not attempted to analyze the transport kinetics of our reporter RNAs. We have rather focused our analysis here on the less mobile molecules that we can confidently identify and analyze (see Materials and methods for details on image acquisition and analysis). Despite the lack of kinetic information, this analysis reflects the behavior of a substantial fraction of the existing RNA population and can provide a valuable characterization of the translation properties of individual RNA molecules in a spatial manner.

To address whether transport to the periphery is accompanied by changes in the translation state of the RNAs, we determined the translation efficiency of single RNA molecules as a function of their distance from the periphery. Consistent with the stochastic translation bursts reported in other systems (*Pichon et al., 2016*; *Wu et al., 2016*; *Yan et al., 2016*), for all three reporters, single RNAs exhibit a range of translation efficiencies (*Figure 4A*). Interestingly, however, plotting the translation efficiency of individual RNAs in relation to their position from the cell edge revealed that translation efficiency is not affected by the distance from the periphery (*Figure 4A*).

Pearson's correlation coefficients in all cases are close to 0 (-0.07 (Rab13), −0.06 (Pkp4), −0.01 (Control)). This result is in agreement with the observations reported above showing that mislocalization of endogenous RNAs does not impact on their translation (*Figures 1* and *2*). Therefore, RNAs trafficked to the periphery through the pathway supported by the Rab13 and Pkp4 UTRs can be translated with similar efficiency in both peripheral and internal locations.

We point out that our analysis is focused on less mobile molecules. We cannot currently assess if translation persists, or not, during periods of active movement. In this regard, we have observed single RNAs that undergo short-range directed movements while being translationally active (*Figure 4—figure supplement 3* and *Videos 5* and *6*). This suggests that these RNAs can be translated while in transit, an observation also noted in other systems (*Wu et al., 2016*). However, due to current limitations on the speed and duration of our imaging, observation of such events is too sporadic to allow conclusions about their frequency.

## APC-dependent RNAs associate with heterogeneous clusters at the tips of protrusions

The data mentioned above have focused on single RNA molecules. However, in the course of our studies we have observed that endogenous APC-dependent RNAs exist in two states: they are found either as single molecules or as clusters

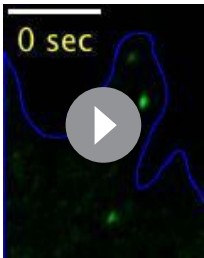

**Video 5.** NIH/3T3 cell expressing scFv-GFP (green), 3x-mCherry-PCP and the localized translation reporter carrying the Rab13 UTR. Images of the GFP channel were acquired sequentially and with no time delay, for the duration of the movie (13 s). Blue line: cell outline. Scale bar: 3 μm. Single frames of this movie are shown in *Figure 4—figure supplement 3A*. The edge of the protrusion is towards the top. One of the observed translation sites moves in an apparently directed manner towards the edge of the protrusion.

DOI: https://doi.org/10.7554/eLife.44752.016

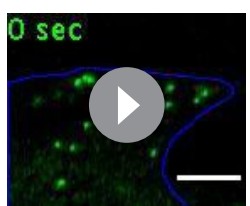

**Video 6.** NIH/3T3 cell expressing scFv-GFP (green), 3x-mCherry-PCP and the localized translation reporter carrying the Pkp4 UTR. Images of the GFP channel were acquired sequentially and with no time delay, for the duration of the movie (19 s). Blue line: cell outline. Scale bar: 3 μm. Single frames of this movie are shown in *Figure 4—figure supplement 3B*. The edge of the protrusion is towards the right. One of the observed translation sites moves in an apparently directed manner towards the edge of the protrusion.
DOI: https://doi.org/10.7554/eLife.44752.017

made up of multiple RNAs in close spatial proximity. We have observed these clusters for virtually all APC-dependent RNAs we have detected by in situ hybridization (*Figure 5* and *Figure 5—figure supplement 1*). These RNA clusters are found at the very tips of cellular protrusions (*Figure 5A–E*). A small fraction of each individual RNA segregates into such clusters, in a 2μm-wide peripheral region (*Figure 5F–H*, *Figure 5—figure supplement 1*), and only a subset of protrusions exhibits such structures (*Figure 5—figure supplement 1*; also see Figure 10 and below for further quantifications). When they form, these clusters are made up of multiple RNA molecules, as assessed by quantitation of FISH signal intensities (*Figure 5A–C* and *Figure 5—figure supplement 1*) and can contain different RNA species. Indeed, two distinct RNAs can be observed in the same RNA cluster (*Figure 5D*). Furthermore, most peripheral clusters of individual RNA species overlap with areas of accumulation of polyadenylated RNA, detected through oligo-dT hybridization (*Figure 5E,I*). Given that the individual detected RNAs (such as the Pkp4 RNA shown in *Figure 5E*) exist in relatively few copies per cell (around, or less than, a hundred copies per cell, *Figure 5—figure supplement 2*), this suggests that the visible accumulation of polyA RNA likely reflects the existence of additional RNA species at that location. Therefore, APC-dependent RNAs are found in heterogeneous RNA clusters at the tips of some protrusions.

This spatially-defined clustering behavior is not observed by RNAs that show a more uniform distribution in the cell body and are not targeted to protrusions (*Figure 5G,H* and *Figure 5—figure supplement 1*; compare APC-dependent RNAs to Arpc3 and P4hb RNAs). Furthermore, this clustering behavior is recapitulated by exogenous localized reporter RNAs. We had previously used reporter RNAs, which carry a series of MS2-binding sites for visualization in the presence of GFP-MS2 coat protein. Reporters carrying 3'UTRs of APC-dependent RNAs form clusters at protrusions whereas a reporter with a control 3'UTR does not (*Mili et al., 2008*). With improved imaging systems we have now used these reporters for higher resolution visualization. MS2-tagged reporter constructs that carry the Rab13 or Net1 3'UTRs, are localized at protrusions, where they can be observed either as individual RNA particles or as clusters of particles at the tips of protrusions (*Figure 5—figure supplement 3*). Furthermore, time lapse imaging of MS2-labeled localized reporters reveals that smaller RNA particles are being trafficked towards and become incorporated into the larger peripheral clusters, further supporting the conclusion that these clusters are made up of multiple RNAs (*Videos 7* and *8*). Therefore overall, transport of APC-dependent RNAs to the periphery can be followed by clustering at the tips of protrusions and this behavior is mediated by signals within the 3'UTRs.

## RNA clusters at the tips of protrusions are translationally silent

To investigate the translation status of RNAs within these clusters we employed the localized translation reporters. Consistent with the data described above, translation reporters carrying

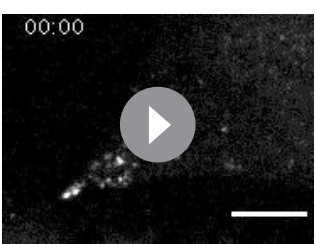

**Video 7.** NIH/3T3 cell expressing tdMCP-GFP (green) and a MS2-reporter RNA carrying the Rab13 UTR. Images of the GFP channel were acquired sequentially and with no time delay, for the duration of the movie (29 s). A single frame of this movie is shown in *Figure 5—figure supplement 3*. Fainter spots reflect single RNA molecules. At the tip of the protrusion a brighter cluster of RNAs is observed. Single particles can be observed moving towards and incorporating into the cluster at the tip (at frames around seconds 3–4 and second 21). Scale bar: 5 μm.
DOI: https://doi.org/10.7554/eLife.44752.022

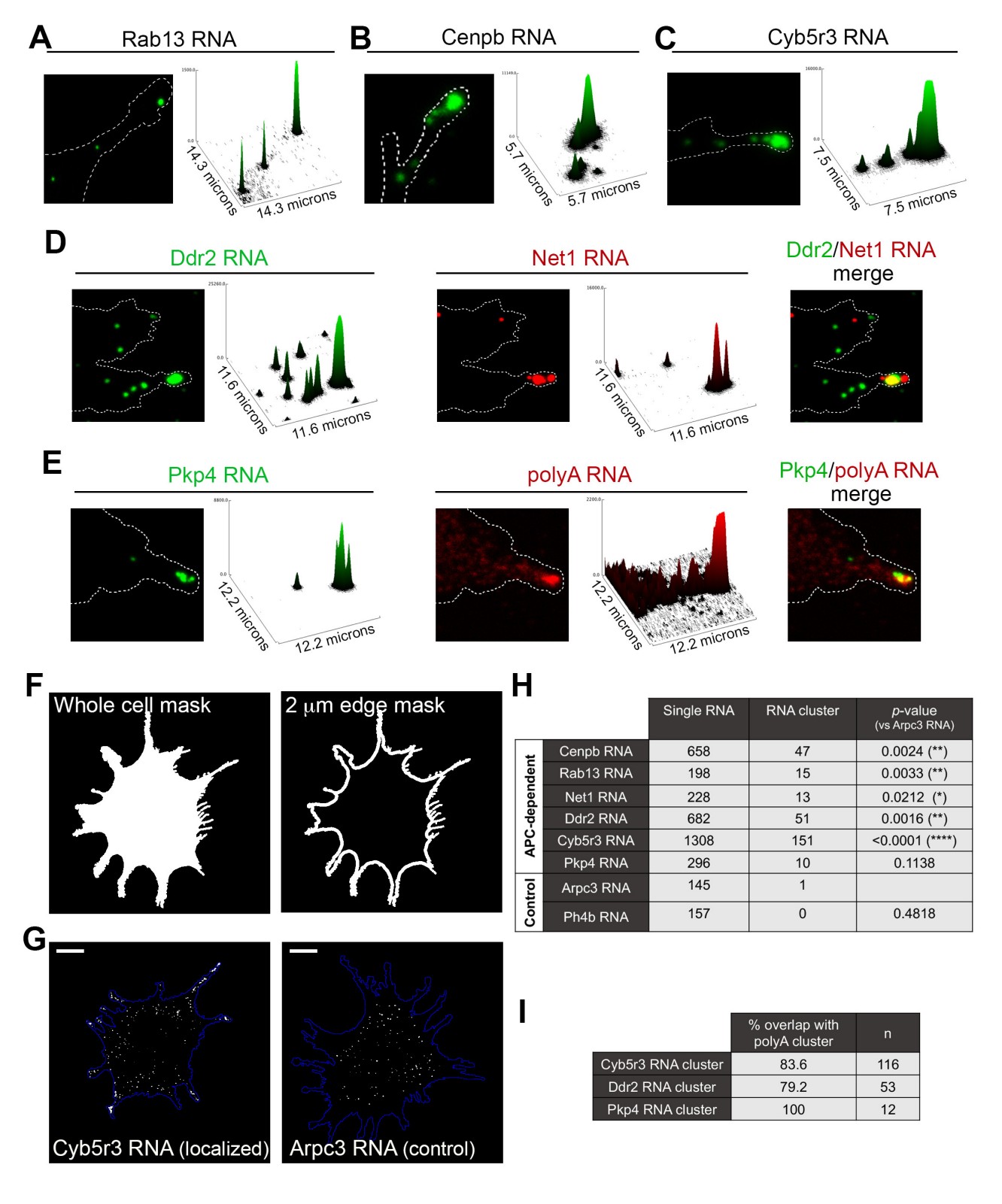

| | | Single RNA | RNA cluster | *p*-value (vs Arpc3 RNA) |
|---|---|---|---|---|
| APC-dependent | Cenpb RNA | 658 | 47 | 0.0024 (**) |
| | Rab13 RNA | 198 | 15 | 0.0033 (**) |
| | Net1 RNA | 228 | 13 | 0.0212 (*) |
| | Ddr2 RNA | 682 | 51 | 0.0016 (**) |
| | Cyb5r3 RNA | 1308 | 151 | <0.0001 (****) |
| | Pkp4 RNA | 296 | 10 | 0.1138 |
| Control | Arpc3 RNA | 145 | 1 | |
| | Ph4b RNA | 157 | 0 | 0.4818 |

| | % overlap with polyA cluster | n |
|---|---|---|
| Cyb5r3 RNA cluster | 83.6 | 116 |
| Ddr2 RNA cluster | 79.2 | 53 |
| Pkp4 RNA cluster | 100 | 12 |

**Figure 5.** APC-dependent RNAs associate with heterogeneous clusters at the tips of protrusions. (**A-C**) The indicated endogenous RNAs were detected by in situ hybridization. Signal intensities of observed spots are shown in the associated surface plot profiles, which also indicate the size of each image in microns. In internal regions all detected RNAs exist as single molecules. At the tips of protrusions, they exist in clusters of multiple RNAs. (**D**) In situ hybridization images and surface plot profiles of endogenous Ddr2 and Net1 RNAs detected in the same cell. Peripheral clusters can contain distinct

*Figure 5 continued on next page*

*Figure 5 continued*

RNA species. (E) In situ hybridization images and surface plot profiles of endogenous Pkp4 RNA and polyA RNA detected in the same cell. Peripheral clusters are characterized by a visible accumulation of polyA RNA. (Note that only enlarged views of individual protrusions are shown in panels A-E). (F) Whole cell masks of cells processed for FISH were used to derive a 2 µm-wide peripheral edge mask. (G) Whole-cell FISH images of the indicated endogenous RNAs (for additional examples see *Figure 5—figure supplement 1*). Scale bars: 15 µm. (H) For each RNA, signal intensity histograms of all detected particles found within the 2µm-wide peripheral edge area, were used to group particles into single RNAs or RNA clusters (see *Figure 5—figure supplement 1* ). Table lists number of particles in each category for the indicated RNAs. p-values based on Fisher's exact test against Arpc3 RNA. (I) Percent of overlap of the indicated RNA clusters with polyA clusters. n = number of particles observed in ca. 25 cells.

DOI: https://doi.org/10.7554/eLife.44752.018

The following figure supplements are available for figure 5:

**Figure supplement 1.** Intensity histograms of endogenous APC-dependent or control RNAs.
DOI: https://doi.org/10.7554/eLife.44752.019

**Figure supplement 2.** Amount of APC-dependent RNAs per cell.
DOI: https://doi.org/10.7554/eLife.44752.020

**Figure supplement 3.** Peripheral cluster formation by MS2-reporter RNAs.
DOI: https://doi.org/10.7554/eLife.44752.021

---

the Rab13 or Pkp4 UTRs could be found at the tips of protrusions in clusters containing multiple RNA molecules, as indicated by the increased mCherry intensity compared to the intensity exhibited by the single molecules in more internal regions (*Figure 6A,B*). Translation reporter clusters were not as pronounced as those observed for endogenous RNAs, likely because of the brief induction time. Nevertheless, strikingly, measuring the translational efficiency of localized reporter RNAs found in clusters revealed that for both reporters, carrying either the Rab13 or Pkp4 UTR, peripherally clustered RNAs are translationally silent (*Figure 6A,B*). Taken together all the above data suggest that, while APC-dependent RNAs are enriched in the periphery, they are translated with similar efficiency in both internal and peripheral locations. At the same time, a subset of them at the tips of protrusions coalesces into clusters which are translationally silent.

## Endogenous Rab13 RNA is translated in both internal and peripheral locations

Our initial expectation was that RNAs at the periphery would be locally translated. While the above data show that this indeed happens, the additional observation of silencing at the periphery was rather counterintuitive. We thus first sought to validate that the above findings indeed reflect the regulation of endogenous localized transcripts. For this, we focused on Rab13 and employed the puro-PLA assay (puromycylation followed by proximity ligation amplification) to detect nascent Rab13 protein molecules in situ (*tom Dieck et al., 2015*). Puro-PLA relies on a brief (5 min) pulse of puromycin to label protein molecules that are being synthesized on ribosomes during the pulse period. Nascent Rab13 can then be visualized by PLA detection of the proximity between an anti-puromycin and an anti-Rab13 antibody (*Figure 7A*). Since puromycin causes chain termination and eventual release of nascent chains, even with a short pulse, it is conceivable that some of the detected signal might reflect released protein that has diffused away from the translation sites. This can be minimized by pretreatment with cycloheximide (CHX) which stalls the nascent proteins on the ribosomes without affecting puromycylation (*David et al., 2012*; *tom Dieck et al., 2015*),

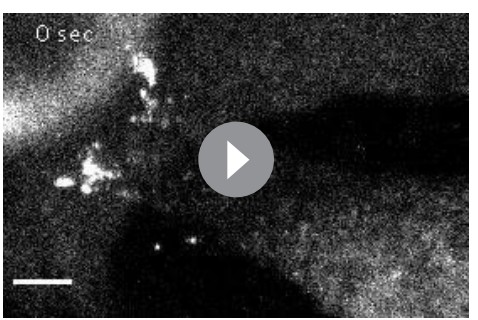

**Video 8.** NIH/3T3 cell expressing tdMCP-GFP (green) and a MS2-reporter RNA carrying the Net1 UTR. Images of the GFP channel were acquired sequentially and with no time delay, for the duration of the movie (22 s). A single frame of this movie is shown in *Figure 5—figure supplement 3*. Fainter spots reflect single RNA molecules. At the edges of the protrusion multiple brighter clusters of RNAs are observed. Single particles from internal regions can be observed moving towards and incorporating into clusters at the tip. Scale bar: 5 µm.

DOI: https://doi.org/10.7554/eLife.44752.023

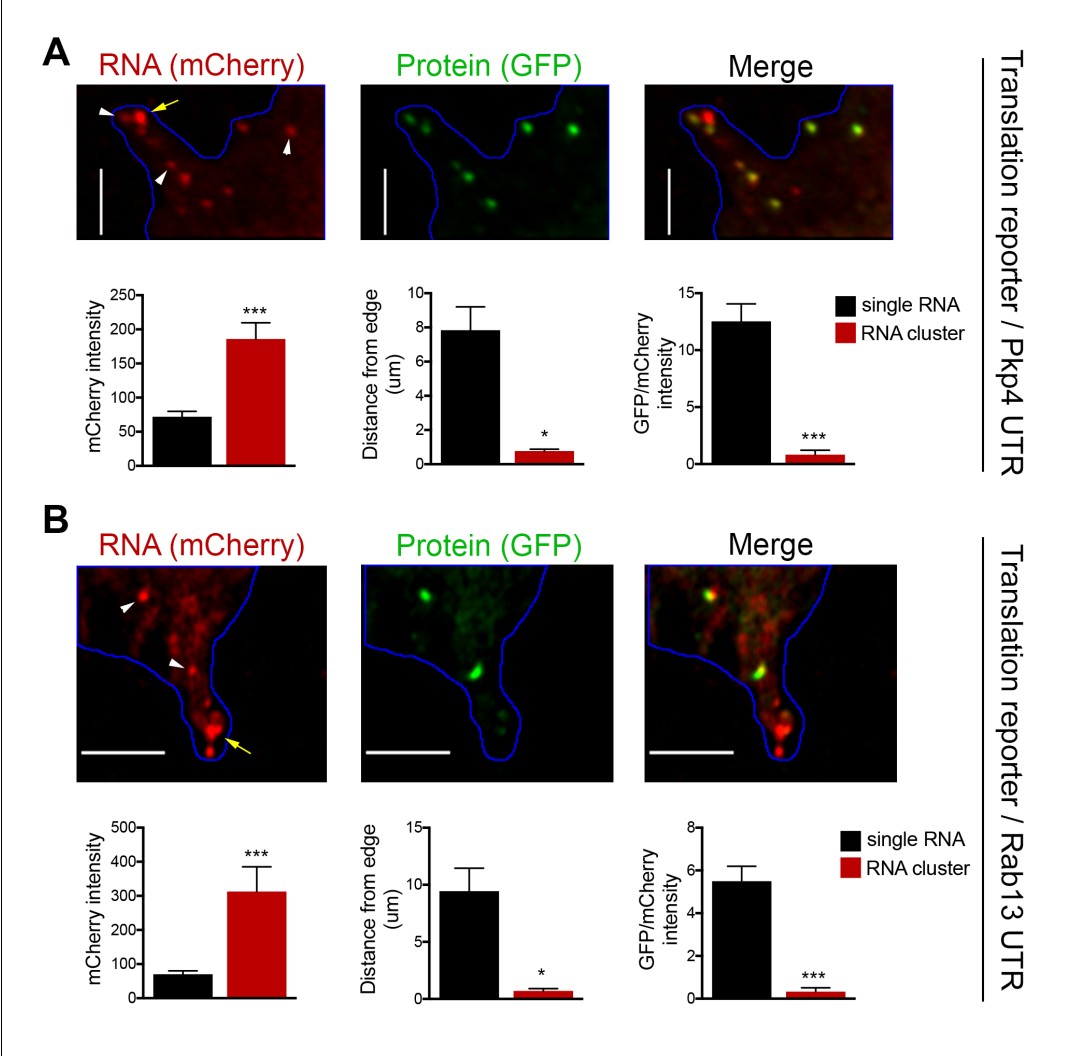

**Figure 6.** RNA clusters at the tips of protrusions are translationally silent. (**A, B**) Imaging of cells expressing localized translation reporters carrying either the Pkp4 (**A**) or Rab13 (**B**) UTRs. White arrowheads point to single RNA molecules. Yellow arrows point to clustered RNAs at the tips of protrusions. mCherry intensity, distance from the edge and GFP/mCherry intensity are plotted for either single RNAs or RNA clusters observed in the same protrusions. error bars: standard error; n = 10 for RNA clusters, n > 25 for single RNAs, from 4 or eight different cells; p-value: *<0.02, ***<0.001 by Student's t-test. Scale bars: 5 μm.

DOI: https://doi.org/10.7554/eLife.44752.024

thus promoting detection of in situ translation sites (*Figure 7A*). The specificity of the signal is validated in cells knocked down for the detected protein as well as in cells pretreated with the translation inhibitors anisomycin or harringtonine, which prevent puromycylation by interfering with the peptidyl-transferase activity or through ribosome run-off respectively. We note that we could not perform these experiments in NIH/3T3 mouse fibroblast cells, because even though the Rab13-puro PLA signal was dependent on translation (i.e. was reduced upon anisomycin treatment), it was not significantly reduced upon Rab13 knockdown, indicating that it mostly originated from non-specific binding of the Rab13 antibody in mouse cells (not shown). We thus used primary human dermal fibroblasts, in which Rab13 protein can be readily detected (*Figure 7—figure supplement 1*). Indeed, Rab13-puro PLA particles in these cells were significantly reduced upon both translation inhibition (with anisomycin or harringtonine) as well as Rab13 knockdown (*Figure 7B*). (Note that the majority of the remaining non-specific signal is concentrated around the nucleus). Furthermore, comparison of the Rab13-puro PLA signal between control and CHX-pretreated cells showed a small but consistent increase in control cells (*Figure 7B*). We interpret this increased number of particles in

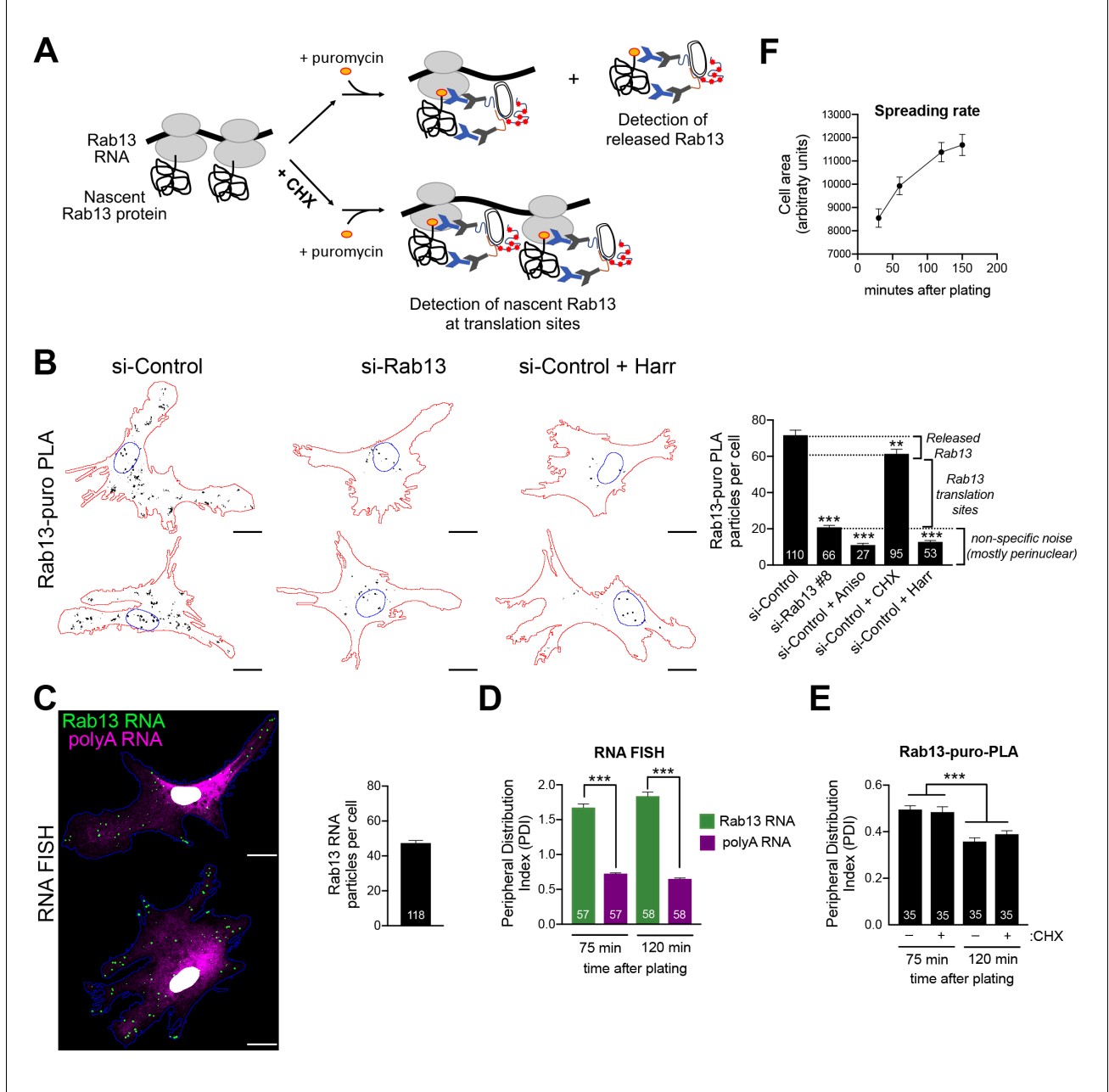

**Figure 7.** Endogenous Rab13 RNA is translated in both internal and peripheral locations, and is silenced at the periphery. (A) Schematic depicting nascent Rab13 protein detection through puro-PLA. Puromycylation leads to detection of both Rab13 released from ribosomes as well as nascent Rab13 at translation sites. Pre-treatment with cycloheximide (CHX) prevents release of nascent protein. (B) Rab13-puro-PLA signal in primary human dermal fibroblasts transfected with control siRNAs, or siRNAs against Rab13, or pre-treated for 15 min with anisomycin (Aniso), cycloheximide (CHX) or harringtonine (Harr). Representative images from some of the conditions are shown on the left and quantitations in the graph. (C) In situ hybridization of Rab13 and polyadenylated (polyA) RNA in primary dermal fibroblasts. Graph shows the average number of Rab13 RNA particles detected per cell. (D, E) Images as those shown in (C) and (B) respectively were used to quantify a peripheral distribution index (PDI) at different times after plating on fibronectin. (F) Cell area of dermal fibroblasts at various timepoints after plating on fibronectin. Error bars: standard error. Number of cells analyzed in 2–4 independent experiments are shown within each bar. For (F) > 145 cells were analyzed for each timepoint. p-value: **<0.01, ***<0.001 by one-way ANOVA with Dunnett's multiple comparisons test, compared to control or indicated samples. Scale bars: 15 μm.

DOI: https://doi.org/10.7554/eLife.44752.025

The following figure supplement is available for figure 7:

**Figure supplement 1.** Rab13 protein levels.

DOI: https://doi.org/10.7554/eLife.44752.026

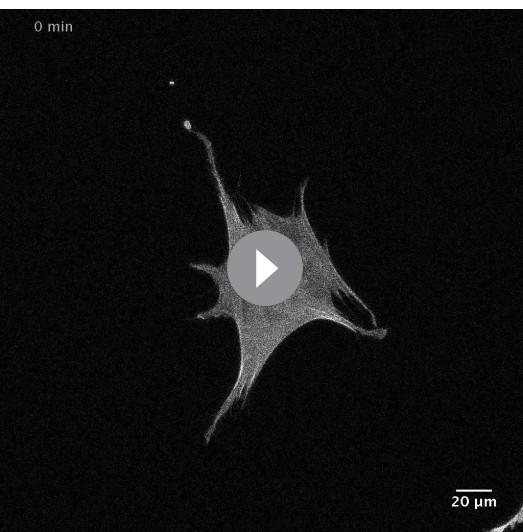

**Video 9.** Primary human dermal fibroblast expressing Lifeact-GFP. Images were acquired every 6 min for a total of 3 hr. Note that each protrusion is very dynamic undergoing retracting and extending phases within minutes.

DOI: https://doi.org/10.7554/eLife.44752.027

control cells to reflect the additional detection of a small amount of nascent Rab13 that is released and diffused away from translation sites. Inferring from these measurements, there are ca. 40 Rab13 translation sites per cell. This number is in good agreement with the ca. 40–50 Rab13 RNA molecules detected by FISH in these cells (*Figure 7C*). We conclude that we can specifically detect nascent Rab13 protein in human fibroblasts and that a large fraction of Rab13 RNAs are being actively translated at any moment.

Consistent with our observations in mouse NIH/3T3 cells, primary human fibroblasts also significantly localize the Rab13 RNA to the periphery compared to the overall distribution of polyadenylated RNAs in the cytoplasm (*Figure 7C,D*). Visual inspection of Rab13 translation sites (Rab13-puro PLA signal) indicated that Rab13 translation occurs in both peripheral and perinuclear locations, consistent with the conclusions reached above. Since we cannot concomitantly detect Rab13 RNAs and Rab13 translation, we measured a peripheral distribution index (PDI) to describe the overall distribution in multiple cells (*Stueland et al., 2019*; *Wang et al., 2017*). Intriguingly, this revealed that Rab13 translation

signal was less peripheral compared to the Rab13 RNA signal (*Figure 7D,E*). While the perinuclear non-specific puro-PLA noise, mentioned above, might exaggerate this difference, it cannot fully account for it. Furthermore, we noticed that upon increased time of spreading on fibronectin-coated coverslips, even though the Rab13 RNA remained peripheral to the same extent (*Figure 7D*), Rab13 translation signal became even less peripheral (*Figure 7E*). The bias towards internal sites is not due to release and trafficking of newly-synthesized Rab13 away from translation sites, because we see the same values when puromycylation is performed after CHX pre-treatment to block nascent protein release (*Figure 7E*). Therefore, while these results showed that indeed endogenous Rab13 is translated in both internal and peripheral locations, they also accentuated the paradox of silencing RNAs after transporting them to the periphery and showed that this effect is enhanced over time during cell spreading.

## Peripheral Rab13 RNA is silenced at retracting protrusions

To try to understand this paradox we looked at the spreading rate of cells around the time points used for translation site imaging. We noticed that the reduction in peripherally translated Rab13 is associated with a decrease in spreading rate (*Figure 7F*). In other words, peripheral translation is reduced as the cells stop extending. Exploring this idea more in dermal fibroblasts was difficult since protrusions of these cells are very dynamic and at any given moment can be extending, retracting or switching behaviors (*Videos 9* and

**Video 10.** Primary human dermal fibroblast expressing Lifeact-GFP. Images were acquired every 6 min for a total of 3 hr. Note that each protrusion is very dynamic undergoing retracting and extending phases within minutes.

DOI: https://doi.org/10.7554/eLife.44752.028

10), thus not allowing us to unambiguously infer the extending or retracting state of a protrusion in fixed cells, and to correlate it with Rab13 translation.

For this reason, we turned to MDA-MB-231 breast cancer cells. These cells, when migrating on collagen-coated glass, exhibit a characteristic tail which retracts over a few minutes (*Figure 8A* and *Videos 11* and *12*). We define these tails as protruding regions that contain actin stress fibers and do not have detectable cortical ruffles visualized through Lifeact-GFP expression in live cells (or phalloidin staining of fixed cells). Live imaging of multiple cells revealed that, once formed, these regions are consistently retracting (90% of 130 protrusions exhibiting retraction in 51 cells). Therefore, visualization of such protrusions allows us to identify with high confidence, even in fixed cells, areas of cell retraction.

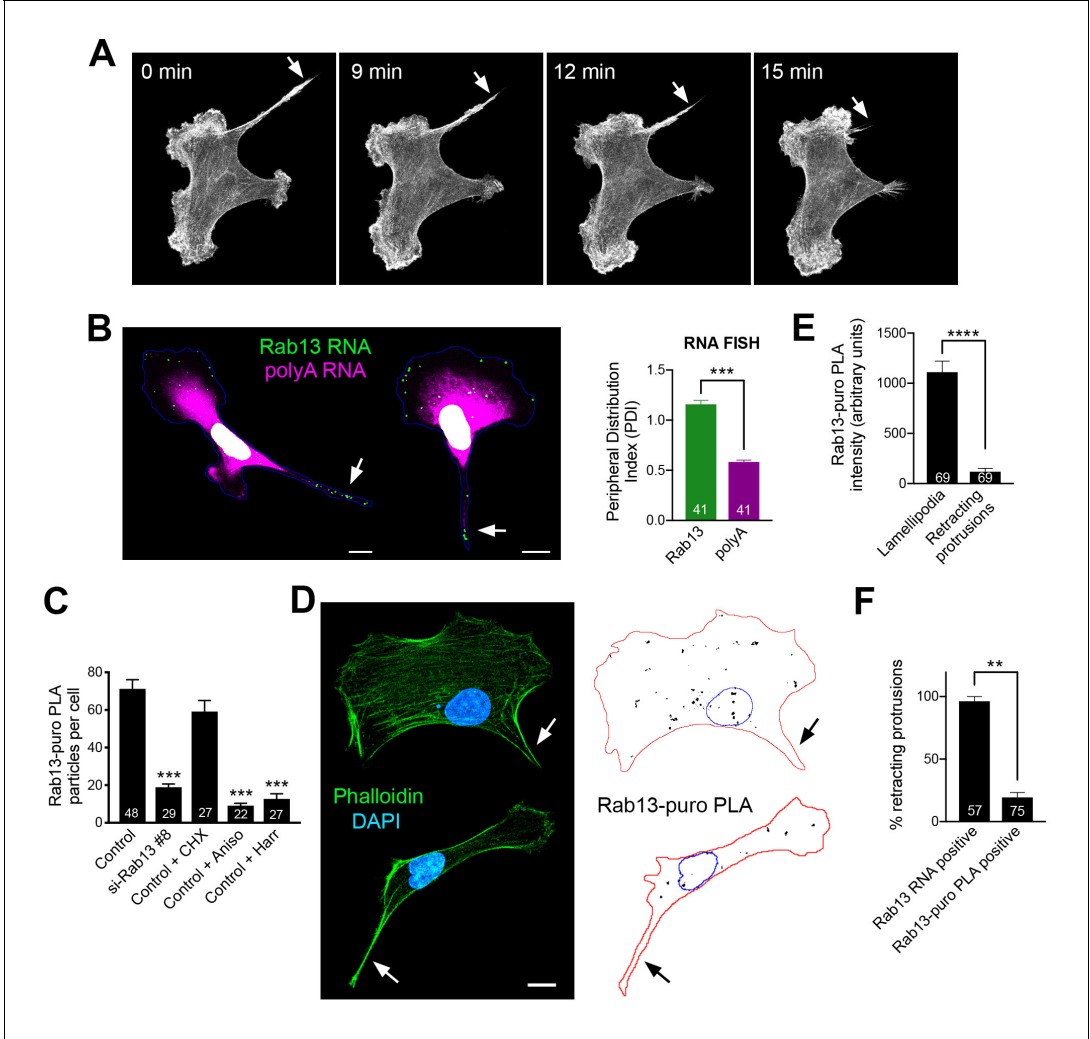

**Figure 8.** Peripheral Rab13 RNA is silenced at retracting protrusions. (A) Snapshots of time lapse imaging of MDA-MB-231 cells expressing Lifeact-GFP. Arrow points to protrusion that retracts within a few minutes. The full-length movie of this sample is presented in *Video 11*. (B) In situ hybridization of Rab13 and polyadenylated (polyA) RNA in MDA-MB-231 cells and PDI quantitations. Arrows point to Rab13 RNA in retracting protrusions. (C) Quantitation of Rab13-puro-PLA signal in MDA-MB-231 cells under the indicated conditions. (D) Representative images of Rab13-puro-PLA and phalloidin staining in MDA-MB-231 cells exhibiting retracting protrusions. Note that Rab13-puro-PLA signal is absent in retracting protrusions (arrows). (E) Rab13-puro-PLA intensity in lamellipodia or retracting protrusions. (See *Figure 9A* for representative outlines). (F) Percent of retracting protrusions positive for Rab13 RNA or puro-PLA signal based on images such as those shown in (B) and (D). Error bars: standard error. Number of cells analyzed in 2–3 independent experiments are shown within each bar. p-value: **<0.01, ***<0.001, ****<0.0001 by Student's t-test (B, E) or one-way ANOVA with Dunnett's multiple comparisons test, compared to control (C). Scale bars: 10 µm.
DOI: https://doi.org/10.7554/eLife.44752.029

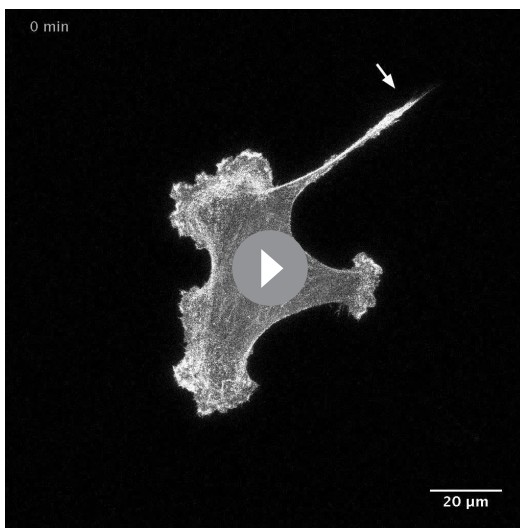

**Video 11.** MDA-MB-231 cell expressing Lifeact-GFP. Images were acquired every 3 min for a total of 51 min. Arrow points to protrusion that retracts over a period of few minutes. Note that lamellipodial regions undergo constant dynamic retracting and extending phases.
DOI: https://doi.org/10.7554/eLife.44752.030

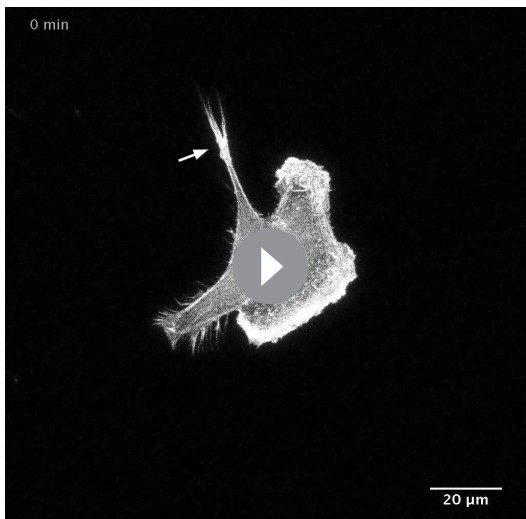

**Video 12.** MDA-MB-231 cell expressing Lifeact-GFP. Images were acquired every 3 min for a total of 51 min. Arrow points to protrusion that retracts over a period of few minutes. Note that lamellipodial regions undergo constant dynamic retracting and extending phases.
DOI: https://doi.org/10.7554/eLife.44752.031

Importantly, in MDA-MB-231 cells, the Rab13 (and other protrusion-localized RNAs) are also peripherally localized and can be found both in front lamellipodia as well as in retracting protrusions (*Figure 8B*). Rab13-puro PLA signal was also readily detected in these cells and was significantly reduced upon both translation inhibition (with anisomycin or harringtonine) as well as Rab13 knockdown (*Figure 8C*). We thus imaged Rab13 RNA, as well as nascent Rab13 protein, in cells exhibiting retracting protrusions (*Figure 8B, D*). We focused our analysis on retracting protrusions and front lamellipodial regions (*Figure 9A*), which despite being quite dynamic exhibit overall net extension (*Videos 11* and *12*). Significantly, Rab13 translation was readily detected in lamellipodia but was drastically reduced in retracting protrusions (*Figure 8E*). This was not due to a difference in Rab13 RNA present in these regions (*Figures 8B* and *9C*). Indeed, while almost all retracting protrusions contained Rab13 RNA, the majority of them were negative for Rab13 translation (*Figure 8F*), strongly indicating that Rab13 RNAs are translationally silenced at retracting protrusions. Therefore, the resolution of the paradox that we propose is that Rab13 and likely other APC-dependent RNAs are translated in extending protrusions/lamellipodia and are silenced upon retraction.

## Silenced Rab13 RNA at retracting protrusions can be found in heterogeneous clusters

In mouse fibroblasts we had observed an association of translationally silent RNAs with peripheral heterogeneous clusters (*Figures 5* and *6*). To determine whether such clustering is also observed for silent RNAs found in retracting protrusions of MDA-MB-231 cells, we imaged the Rab13 RNA and analyzed the observed particles either in translationally-active lamellipodial regions or in the translationally-silent retracting protrusions (*Figure 9A*). A frequency distribution histogram revealed that, in lamellipodia, the majority of Rab13 RNA particles exhibit intensities centered around a single peak indicating that they largely exist as single molecules. By contrast, in retracting protrusions a significant number of particles were larger and exhibited increased intensities indicating that they correspond to clusters of multiple Rab13 RNA molecules (*Figure 9B,C*). Therefore, corroborating our previous observations, translationally silent Rab13 RNA is found in multimeric clusters. Furthermore, since retracting protrusions exhibit a large fraction of single Rab13 RNAs, these

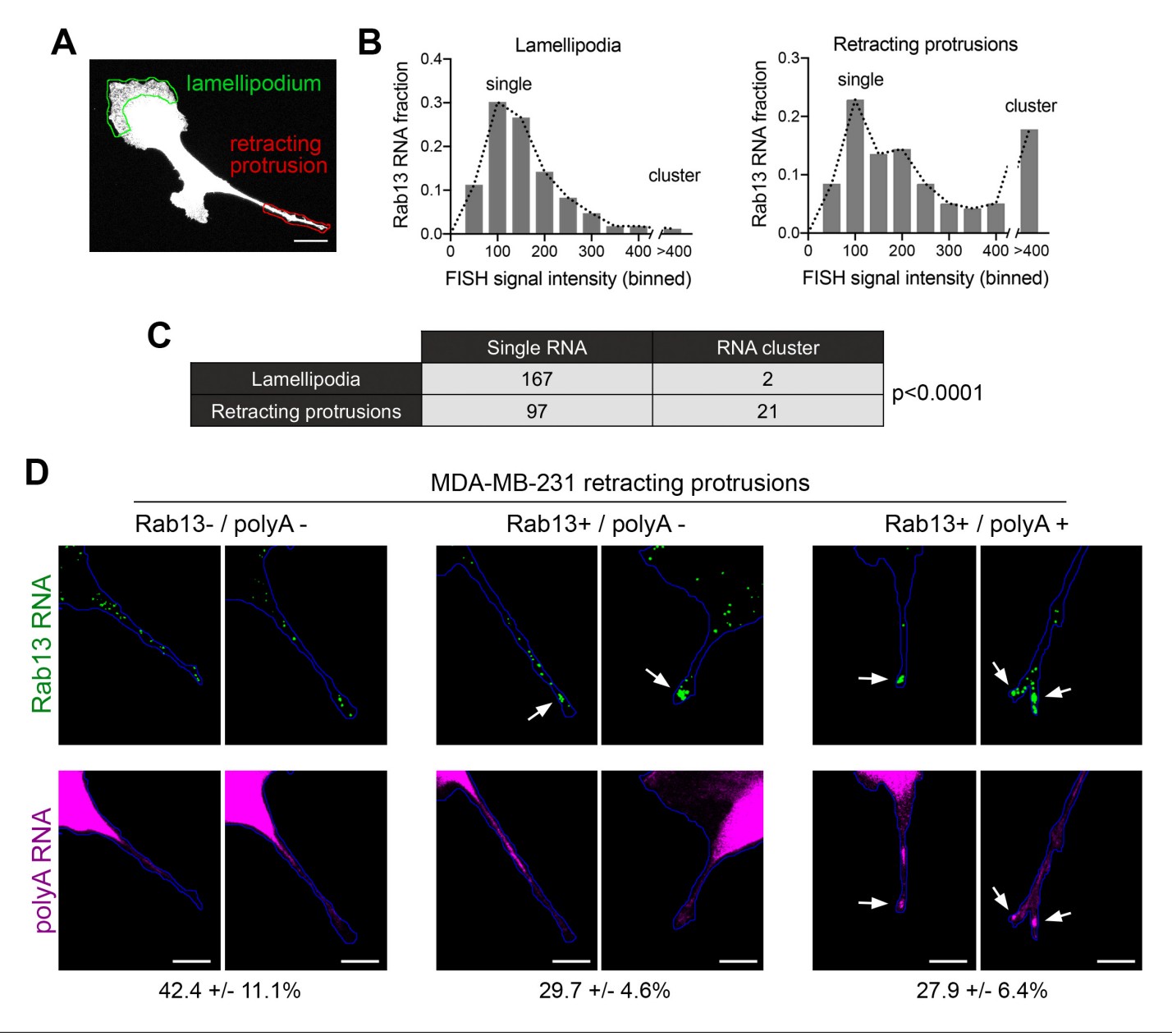

**Figure 9.** Silenced Rab13 RNA at retracting protrusions can be found in heterogeneous clusters. (A) Outlines of 'lamellipodia' or 'retracting protrusion' regions used for quantitations Scale bar:10 µm. (B, C) Frequency distribution histograms of signal intensities (in arbitrary units) of Rab13 RNA particles within lamellipodia or retracting protrusions of MDA-MB-231 cells, as shown in (A). Intensities > 400 were grouped in one bin and indicate RNA clusters. Table lists numbers of single RNAs or RNA clusters observed in 32 cells. p-value by Fisher's exact test. Essentially identical results were obtained in three independent experiments. (D) Retracting MDA-MB-231 protrusions (outlined in blue) stained for Rab13 and polyA RNAs. Based on the staining pattern, protrusions were grouped into three categories: Rab13-/polyA- do not exhibit visible Rab13 clusters or obvious local accumulations of polyA RNA; Rab13+/polyA- exhibit clusters of Rab13 RNA (arrows) but no obvious polyA clusters; Rab13+/polyA +exhibit Rab13 clusters which coincide with obvious corresponding polyA clusters (arrows). Values indicate average fraction of protrusions in each category ± standard error. n = 60 from two independent experiments. Scale bars: 10 µm.

DOI: https://doi.org/10.7554/eLife.44752.032

results additionally indicate that containment within clusters is not required for silencing but might be a consequence of it.

To determine whether RNA clusters in retracting MDA-MB-231 protrusions were composed of heterogeneous RNA species, we tested for the co-localization of Rab13 with polyA RNA. As

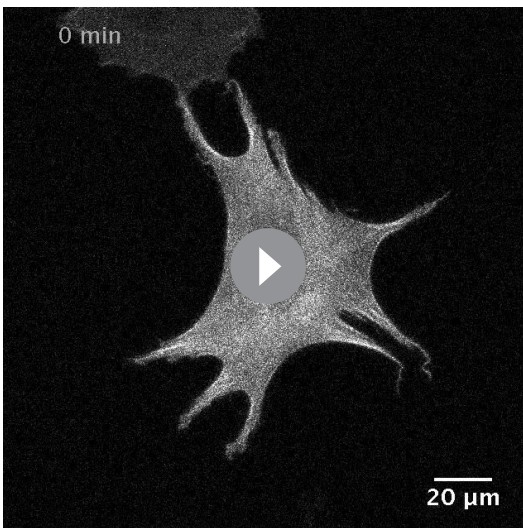

**Video 13.** NIH/3T3 fibroblast expressing Lifeact-GFP. Images were acquired every 6 min for a period of 1 hr. Note that, in comparison to MDA-MB-231 cells, NIH/3T3 cells exhibit much slower dynamics with protrusions persisting relatively unchanged for the duration of imaging.
DOI: https://doi.org/10.7554/eLife.44752.033

mentioned above, in fibroblast protrusions, the majority of individual RNA clusters overlap with areas of visible concentration of polyA RNA (*Figure 5E,I*). We observed something different in MDA-MB-231 cells (*Figure 9D*). Examination of multiple protrusions revealed that $29.7 \pm 4.6\%$ of protrusions (n = 60) contained clusters of Rab13 RNA without any corresponding accumulation of polyA RNA signal, while in $27.9 \pm 6.4\%$ of protrusions Rab13 RNA appeared to be part of more heterogenous, polyA-positive assemblies. Considering that polyA intensity is reflective of the amount of contained RNAs, it therefore appears that MDA-MB-231 cells form a spectrum of clusters of different sizes. Given that the lifetimes of retracting protrusions are noticeably different in the two cell types (with NIH/3T3 protrusions persisting for a longer time, while MDA-MB-231 protrusions quickly retracting within a few minutes; compare *Videos 11* and *12* to *Video 13*), a likely possibility is that the observed differences reflect the dynamics of these granules (see discussion).

## Formation of RNA clusters at protrusions is promoted by translational inhibition and requires microtubules

To probe more into the assembly of these peripheral clusters, we tested how their formation is affected upon translational inhibition or disruption of RNA transport to the periphery. We performed these experiments in mouse fibroblast cells, in which peripheral clusters can be readily detected through polyA accumulation (*Figure 5I*). Indeed, in these cells, the local concentration of polyA RNA at protrusions is a more reliable identifier of peripheral RNA clusters compared to the detection of any one particular RNA species that constitutes them. This is especially true in the case of low-abundance RNAs, such a Pkp4, for which a large proportion of protrusions exhibit visible polyA RNA clusters that either contain single Pkp4 RNA molecules (*Figure 10—figure supplement 1*; white arrows; 45.9% (n = 98)), or are devoid of Pkp4 RNA (*Figure 10—figure supplement 1*; yellow arrowheads; 41.8% (n = 98)). Therefore, at least for 3T3 cells, polyA RNA reveals more accurately the presence of peripheral clusters.

Scoring of cell populations for the presence of polyA granules at protrusions revealed that 25–30% of protrusions contain peripheral polyA granules (*Figure 10*). Interestingly, inhibition of translation doesn't disrupt their formation, contrasting with the behavior exhibited by other types of cytoplasmic RNA granules, such as stress granules, which are rapidly dissolved upon inhibition of translation elongation. Instead, for peripheral polyA granules, brief treatment with cycloheximide or puromycin promotes their appearance (*Figure 10*), suggesting that translational inhibition is a limiting step in the formation of RNA clusters at protrusions.

We further disrupted the microtubule network with nocodazole, or specifically disrupted detyrosinated microtubules using the tubulin carboxypeptidase inhibitor parthenolide (*Figure 10*). Of note, both treatments result in gradual retraction of protrusions and consequently in an increase in the number of cells with rounded morphology where peripheral RNA granules cannot be detected. To discount the possibility that the absence of peripheral granules was a secondary effect of the observed changes in cell shape, we focused our analysis only on cells that maintained cell protrusions and had similar morphology to the control conditions (*Figure 10*). Even looking at this narrower group of cells revealed that disruption of microtubules, or detyrosinated microtubules, significantly reduced the appearance of polyA RNA granules at protrusions (*Figure 10*). These results, together with the data presented above, are consistent with the model that peripheral RNA granules result from, and require, RNA transport to the periphery.

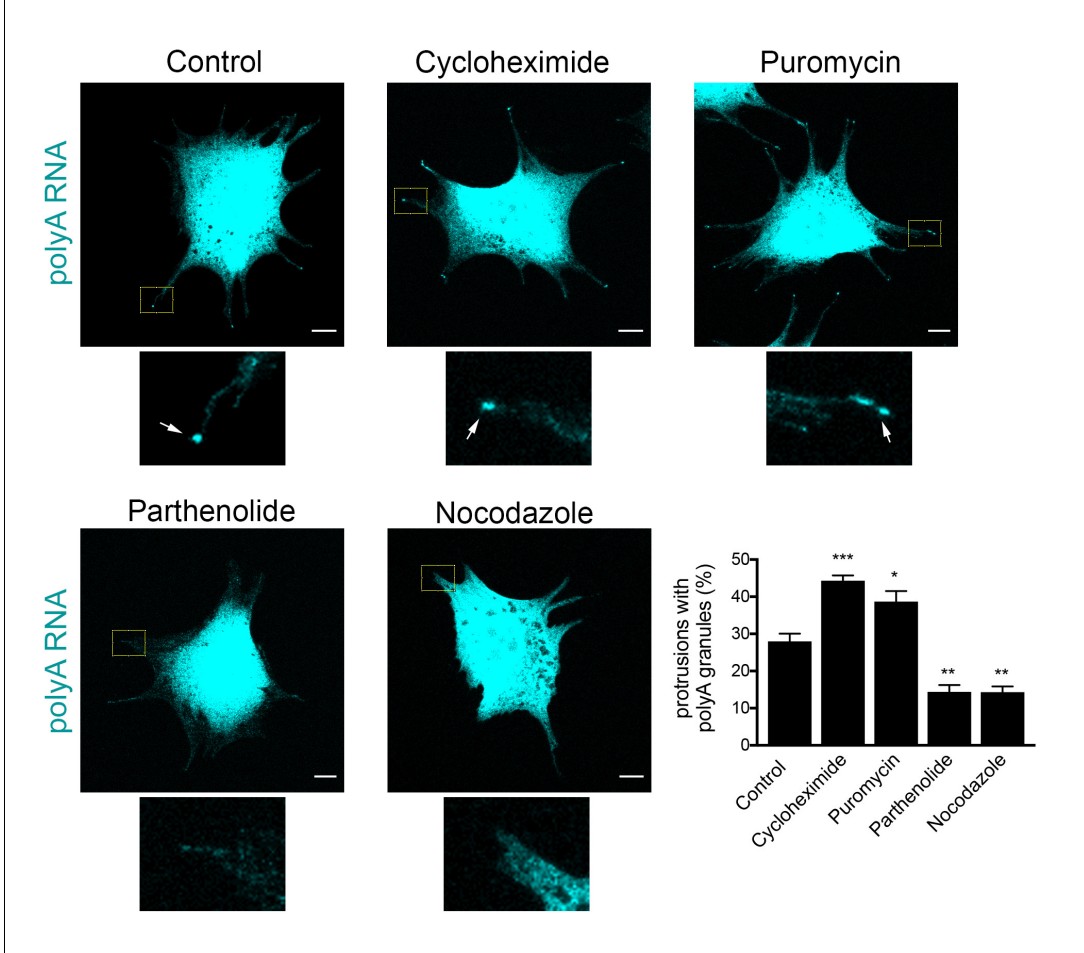

**Figure 10.** Formation of RNA clusters at protrusions is promoted by translational inhibition and requires microtubules. PolyA RNA was detected in NIH/3T3 cells with the indicated treatments. Boxed regions are enlarged to show the presence (arrows) or absence of polyA RNA granules at the tips of protrusions. Graph shows scoring of protrusions for the presence of polyA RNA granules. Values are mean and standard error of at least three independent experiments. For each experiment approximately 300 protrusions from more than 25 cells were observed. p-value: *<0.02, **<0.01, ***<0.001 by one-way ANOVA with Dunnett's multiple comparisons test, compared to control. Scale bars: 10 μm.

DOI: https://doi.org/10.7554/eLife.44752.034

The following figure supplement is available for figure 10:

**Figure supplement 1.** PolyA RNA staining is a more reliable identifier of peripheral clusters in 3T3 cells.

DOI: https://doi.org/10.7554/eLife.44752.035

## Discussion

Here, we investigate the translational regulation of APC-dependent RNAs which are targeted to cell protrusions. We find that the cytoplasmic position of APC-dependent RNAs does not affect their translation, since they can be translated similarly in both internal and peripheral locations. Rather, translation of APC-dependent RNAs is coordinated with specific peripheral cellular processes, being activated at extending protrusions/lamellipodia and suppressed upon protrusion retraction. Silencing is coupled to a change in the physical state of the RNAs manifested as single RNAs clustering into heterogenous granules at the tips of protrusions.

This mode of regulation is distinct from the one proposed for several other localized transcripts, whereby RNAs are transported in a silenced state and are translationally activated only upon reaching the final destination or upon receipt of specific signals (*Besse and Ephrussi, 2008*; *Buxbaum et al., 2015*; *Jung et al., 2014*). A reason for this latter type of regulation has been proposed to be the need to prevent protein appearance at sites, or times, where it might be deleterious. Indeed, premature appearance of the transcriptional repressor Ash1p in the mother cell during

budding suppresses transcriptional programs in both mother and daughter cells and prevents mating-type switching (*Long et al., 1997*). Similarly, disrupting the timing of translational activation within neuronal axons or dendrites can lead to aberrant axonal pathfinding or synaptic responses (*Colak et al., 2013*; *Holt and Schuman, 2013*; *Jung et al., 2012*).

Our observation of translation regardless of cytoplasmic location suggests that the proteins encoded by APC-dependent RNAs can be produced in internal regions without deleterious effects. We propose that this mode of regulation could additionally have functional implications for the encoded proteins. Specifically, translation in local environments can attribute proteins with different properties. This could result from differential protein modifications or through proximity to protein partners, which during co-translational assembly could affect the type or efficiency of multimeric complex formation (*Basu et al., 2011*; *Jung et al., 2014*; *Shiber et al., 2018*; *Shieh et al., 2015*). In light of these ideas, we suggest that a single mRNA, as it is being continuously translated at various stages during its transport to the periphery, could give rise to protein copies which have different properties, and therefore functional potential, depending on the local micro-environment they are translated into.

We additionally show that translation of APC-dependent RNAs is specifically suppressed in retracting protrusions and this silencing is associated with the formation of multimeric heterogeneous clusters. Formation of these clusters is reduced by treatments that prevent RNA transport to the periphery and is increased upon global translational inhibition, suggesting that silencing is a limiting step in their formation. Nevertheless, silencing can occur outside of clusters, since a large proportion of RNAs are observed as single particles in retracting, translationally-silent protrusions. The appearance of these heterogenous RNA clusters at the tips of protrusions, and their relation to translation, are reminiscent of other types of RNA granules formed by liquid-liquid phase separation (*Courchaine et al., 2016*; *Weber and Brangwynne, 2012*). For example, stress granules (SGs) and processing bodies (PBs) are sites of dynamic concentration of RNA molecules, but in both cases translational silencing or decay can occur regardless of RNA localization within granules (*Halstead et al., 2015*; *Panas et al., 2016*; *Perez-Pepe et al., 2018*). SGs and PBs form throughout the cytoplasm under normal conditions or in response to stress. An interesting distinction of the clusters described here is that their formation is induced in particular subcellular regions associated with protrusion retraction, suggesting that their assembly/disassembly is controlled by spatial signals.

The silencing of Rab13, and likely of other APC-dependent RNAs, at retracting protrusions is contrasted by their translational activation in extending lamellipodia. Given that localization of APC-dependent RNAs to the periphery is important for cell migration (*Wang et al., 2017*) these data could point to a functional role for spatially segregating translation such that local protein production occurs in actively extending regions, while being suppressed in retracting areas. This would imply the existence of a dynamic regulatory mechanism that coordinates APC-dependent RNA translation with the continuous protrusion and retraction cycles that characterize cellular migration (*Ji et al., 2008*; *Tkachenko et al., 2011*).

A potential underlying mechanism for local silencing and granule formation could rely on spatially restricted phosphorylation/dephosphorylation events, which have been shown to affect the propensity of RNA-binding proteins to form phase-separated granules or to bind to RNAs (*Monahan et al., 2017*; *Murray et al., 2017*; *Thapar, 2015*). In this regard, it is interesting that proteins associating with APC-dependent RNAs include the translational regulator FMRP and the RNA-binding protein FUS, one of the paradigm proteins used in understanding phase transitions (*Mili et al., 2008*; *Yasuda et al., 2017*; *Yasuda et al., 2013*). Local modifications of FMRP or FUS could underlie the observed regulation. Additional local events, including local maturation of miRNAs, could be envisioned (*Sambandan et al., 2017*).

With regards to the assembly/disassembly dynamics of peripheral clusters and the fate of the sequestered RNAs, limitations in the duration of live-imaging we can accomplish do not permit concrete conclusions. However, our current data indicate that the extent of cluster formation might be influenced by the rate of retraction. Specifically, in NIH/3T3 cells, the majority of peripheral clusters are characterized by a visible accumulation of polyA signal, indicative of a high concentration of heterogeneous RNA species. By contrast, a substantial fraction of retracting MDA-MB-231 protrusions do not exhibit obvious polyA accumulation (*Figure 9*). A prominent difference between the two cell types is the speed with which protrusions retract. Contractile protrusions of NIH/3T3 cells can persist for a long time, while MDA-MB-231 protrusions quickly retract within a few minutes (compare

*Videos 11* and *12* to *Video 13*). While we cannot rule out other interpretations, such as the existence of deadenylated transcripts, we favor the idea that slowly retracting protrusions allow for the build-up of large heterogenous granules, while in faster retracting protrusions peripheral granules are transient, and rapidly disassemble either through RNA degradation or release into the cytosol.

The existence of translationally silent multimeric RNA clusters also offers a potential explanation for the slight decrease of APC-dependent RNAs found in the heavy RNP fraction under conditions that disrupt transport to the periphery (*Figures 1D* and *2B*). Translationally silent, higher-order RNP complexes can sediment at sucrose density gradient fractions heavier than their translated counterparts (*Chekulaeva et al., 2006*). By analogy, the peripheral clusters of APC-dependent RNAs likely account for some of the RNA found at heavier fractions of the gradient. Reduction of their formation upon parthenolide treatment (*Figure 10*) could likely account for the apparent shift of APC-dependent RNAs towards the lighter polysome fraction of the gradient (*Figures 1D* and *2B*).

We had previously reported that APC-dependent RNAs can also be found in internal cytoplasmic granules induced by expression of FUS mutants carrying ALS-associated mutations. Intriguingly, in these FUS-granules APC-dependent RNAs are translationally active (*Yasuda et al., 2013*). Taken together, these observations raise the possibility that, at least for APC-dependent RNAs, their physical partitioning into granules is not the sole determinant, but acts in combination with the particular local environment to determine the eventual impact on their translation status. It would be interesting to further investigate how the composition of peripheral polyA granules differs from other internal cytoplasmic RNA granules with regards to both RNA and protein constituents.

We note that the above study describes the regulation of APC-dependent RNAs in migrating mesenchymal cells. It would be interesting to explore how translation and transport of this RNA group is carried out in larger and more stably polarized cells such as neurons.

Overall, we describe here a distinct mode of translational regulation of localized RNAs. These findings provide a different perspective towards understanding how local translation can influence protein activities and how these regulatory mechanisms could be integrated with dynamic cellular behaviors.

# Materials and methods

**Key resources table**

| Reagent type (species) or resource | Designation | Source or reference | Identifiers | Additional information |
|---|---|---|---|---|
| Cell line (*Homo sapiens*) | MDA-MB-231 | ATCC | ATCC Cat# HTB-26, RRID: CVCL_0062 | |
| Cell line (*Mus musculus*) | NIH/3T3 | ATCC | RRID:CVCL_0594 | |
| Cell line (*Homo sapiens*) | Primary dermal fibroblasts | provided by Dr. Ramiro Iglesias-Bartolome (NCI, NIH) | | |
| Cell line (*Mus musculus*) | NIH/3T3-PCFvGdelta1-PP7 (control translation reporter) | This study | | NIH/3T3 cells expressing SunTag-based control translation reporter. |
| Cell line (*Mus musculus*) | NIH/3T3-PCFvGdelta1-deltaPP7 (control translation reporter) | This study | | NIH/3T3 cells expressing SunTag-based control translation reporter with deletion of the PP7 repeats. |
| Cell line (*Mus musculus*) | NIH/3T3-PCFvGdelta1-Rab13U (translation reporter/Rab13 UTR) | This study | | NIH/3T3 cells expressing SunTag-based translation reporter with the mouse Rab13 UTR. |

*Continued on next page*

*Continued*

| Reagent type (species) or resource | Designation | Source or reference | Identifiers | Additional information |
|---|---|---|---|---|
| Cell line (*Mus musculus*) | NIH/3T3-PCFvGdelta1-Pkp4U (translation reporter/Pkp4 UTR) | This study | | NIH/3T3 cells expressing SunTag-based translation reporter with the mouse Pkp4 UTR. |
| Cell line (*Mus musculus*) | NIH/3T3-tdMCP-GFP_pIND20-b24bs/Net1 (MS2 reporter/Net1 UTR) | This study | | NIH/3T3 cells expressing MS2 reporter with the mouse Net1 UTR. |
| Cell line (*Mus musculus*) | NIH/3T3-tdMCP-GFP_pIND20-b24bs/Rab13 (MS2 reporter/Rab13 UTR) | This study | | NIH/3T3 cells expressing MS2 reporter with the mouse Rab13 UTR. |
| Antibody | anti-Rab13 rabbit polyclonal | Novus Biologicals | NBP1-85799 | (1:1,000 WB; 1:200 PLA) |
| Antibody | anti-GAPDH (14C10) rabbit monoclonal | Cell Signaling | 2118 | (1:2,000 WB) |
| Antibody | anti-puromycin (3RH11) mouse monoclonal | Kerafast | EQ0001 | (1:2,000 PLA) |
| Recombinant DNA reagent | pHR-tdPP7-3x-mCherry | Addgene | 74926 | |
| Recombinant DNA reagent | pcDNA4TO-24x GCN4_v4-kif18b-24xPP7 | Addgene | 74928 | |
| Recombinant DNA reagent | pcDNA4TO-24x GCN4_v4-kif18b | Addgene | 74934 | |
| Recombinant DNA reagent | pHR-scFv-GCN4-sfGFP-GB1-NLS-dWPRE | Addgene | 60906 | |
| Recombinant DNA reagent | pHR-scFv-GCN4-sfGFP-GB1-deltaNLS-dWPRE | This study | | Plasmid expressing scFv against GCN4 peptide of the SunTag system, fused to sfGFP, without NLS. Derived from pHR-scFv-GCN4-sfGFP-GB1-NLS-dWPRE after introduction of stop codon before the NLS sequence. |
| rRcombinant DNA reagent | pInducer 20 | Addgene | 44012 | |
| Recombinant DNA reagent | Phage-ubc-nls-ha-tdMCP-gfp | Addgene | 40649 | |
| Recombinant DNA reagent | pInducer20-24xGCN4_v4-kif18b-24xPP7-Rab13 UTR | This study | | Dox-inducible translation reporter carrying the mouse Rab13 3'UTR |
| Recombinant DNA reagent | pInducer20-24xGCN4_v4-kif18b-24xPP7-Pkp4 UTR | This study | | Dox-inducible translation reporter carrying the mouse Pkp4 3'UTR |
| Recombinant DNA reagent | pInducer20-beta 24bs-Rab13 UTR | This study | | Dox-inducible MS2 reporter carrying the mouse Rab13 3'UTR |
| Recombinant DNA reagent | pInducer20-beta 24bs-Net1 UTR | This study | | Dox-inducible MS2 reporter carrying the mouse Net1 3'UTR |
| Sequence-based reagent | All Stars Negative control siRNA | Qiagen | 1027281 | |

*Continued on next page*

*Continued*

| Reagent type (species) or resource | Designation | Source or reference | Identifiers | Additional information |
|---|---|---|---|---|
| Sequence-based reagent | Rab13 siRNA, si-Rab13 #8 | Qiagen | SI02662702 | target sequence: 5'-ATGGTCTTTCTT GGTATTAAA-3' |
| Sequence-based reagent | FISH probes against mouse Net1 | Thermo Fisher Scientific | VB1-3034209 | |
| Sequence-based reagent | FISH probes against mouse Cyb5r3 | Thermo Fisher Scientific | VB1-18647 | |
| Sequence-based reagent | FISH probes against mouse Cenpb | Thermo Fisher Scientific | VB1-18648 | |
| Sequence-based reagent | FISH probes against mouse Rab13 | Thermo Fisher Scientific | VB1-14374 | |
| Sequence-based reagent | FISH probes against human Rab13 | Thermo Fisher Scientific | VA1-12225 | |
| Sequence-based reagent | FISH probes against mouse Pkp4 | Thermo Fisher Scientific | VB4-600264 | |
| Sequence-based reagent | FISH probes against human Kif18b | Thermo Fisher Scientific | VA6-3170686 | |
| Sequence-based reagent | FISH probes against mouse Ddr2 | Thermo Fisher Scientific | VB1-14375 | |
| Sequence-based reagent | FISH probes against mouse Arpc3 | Thermo Fisher Scientific | VB1-14507 | |
| Sequence-based reagent | FISH probes against mouse P4hb | Thermo Fisher Scientific | VB6-15898 | |
| Sequence-based reagent | Custom-made codeset | NanoString Technologies | Item # 116000002 | |
| Commercial assay or kit | Duolink In Situ Red kit | Sigma Aldrich | DUO92101 | |
| Commercial assay or kit | ViewRNA ISH Cell Assay kit | Thermo Fisher Scientific | QVC0001 | |

## Plasmid constructs and lentivirus production

Plasmids for translation reporters: pHR-tdPP7-3x-mCherry, pcDNA4TO-24xGCN4_v4-kif18b-24xPP7 and pcDNA4TO-24xGCN4_v4-kif18b were gifts from Marvin Tanenbaum (Addgene plasmids #74926, 74928 and 74934 respectively). pcDNA4TO-24xGCN4_v4-kif18b-24xPP7 was used to introduce different mouse UTR sequences at EcoRI/AscI sites after the PP7 repeats, and the inserts encompassing the coding sequence and UTRs were transferred into pInducer 20 lentivector (gift of Stephen Elledge, Addgene plasmid # 44012), using the Gateway LR clonase II Enzyme mix (Thermo Fisher Scientific, cat# 11791–020) according to the manufacturer's instructions, to generate plasmids: pInducer20-24xGCN4_v4-kif18b-24xPP7-Rab13 UTR and pInducer20-24xGCN4_v4-kif18b-24xPP7-Pkp4 UTR. pHR-scFv-GCN4-sfGFP-GB1-NLS-dWPRE was a gift from Ron Vale (Addgene plasmid # 60906). The NLS sequence in this construct was deleted and replaced with a stop codon, to generate pHR-scFv-GCN4-sfGFP-GB1-deltaNLS-dWPRE.

Plasmids for MS2-GFP-reporters: Phage-ubc-nls-ha-tdMCP-gfp was a gift from Robert Singer (Addgene plasmid # 40649). pcDNA3-based plasmids expressing the β-globin genomic sequence followed by 24xMS2 binding sites and different 3'UTRs, were previously described (*Mili et al., 2008*). Inserts from these constructs were transferred into pInducer 20 lentivector for inducible expression, to generate plasmids pInducer20-beta24bs-Rab13UTR and pInducer20-beta24bs-Net1UTR. mEGFP-Lifeact-7 (gift of Michael Davidson; Addgene plasmid # 54610) was transferred into pCDH-CMV-MCS-EF1-Puro (System Biosciences, cat #CD510B-1) using NheI/NotI sites for virus production.

Lentiviruses were produced in HEK293T cells cultured in DMEM containing 10% FBS and Penicillin/Streptomycin. HEK293T cells were transfected with lentivectors, together with packaging

plasmids pMD2.G and psPAX2 using PolyJet In Vitro DNA transfection Reagent (SignaGen) for 48 hr. Harvested virus was precipitated with Polyethylene Glycol at 4 °C overnight.

## Western blot

For Western blot detection the following antibodies were used: anti-Rab13 rabbit polyclonal (Novus Biologicals, cat# NBP1-85799) and anti-GAPDH rabbit monoclonal 14C10 (Cell Signaling Technology, cat# 2118).

## Cell culture and generation of cell lines

NIH/3T3 mouse fibroblast cells (ATCC) were grown in DMEM supplemented with 10% calf serum, sodium pyruvate and penicillin/streptomycin (Invitrogen) at 37°C, 5% $CO_2$. The stable NIH/3T3 cell line that inducibly expresses the Pkp4 cUTR has been described before (*Wang et al., 2017*). Primary human dermal fibroblasts were kindly provided by Dr. Ramiro Iglesias-Bartolome (Center for Cancer Research, NCI, NIH) and were cultured in DMEM supplemented with 10% fetal bovine serum, sodium pyruvate and penicillin/streptomycin (Invitrogen) at 37°C, 5% $CO_2$. MDA-MB-231 breast cancer cells (ATCC) were grown in Leibovitz's L15 media supplemented with 10% fetal bovine serum and penicillin/streptomycin at 37°C in atmospheric air. Cell lines have been tested for mycoplasma and are free of contamination. To generate cell lines expressing translation reporters, NIH/3T3 cells were sequentially infected with lentiviruses expressing tdPP7-3x-mCherry and scFv-GFP; a subpopulation was isolated through fluorescence activated cell sorting (FACS) and clonal lines were derived. A line expressing uniformly low levels of mCherry and GFP was used to introduce the various pInducer 20-based reporter constructs, and stably expressing cells were selected with geneticin (Thermo Fisher Scientific). The derived cell lines are: PCFvGdelta1-PP7 (control translation reporter); PCFvGdelta1-deltaPP7 (control translation reporter with deletion of the PP7 repeats); PCFvGdelta1-Rab13U (translation reporter with Rab13 UTR); PCFvGdelta1-Pkp4U (translation reporter with Pkp4 UTR). Expression of the reporters was induced by addition of 1 μg/ml Doxycycline (Fisher Scientific) approximately 2–3 hr before imaging.

To generate cell lines expressing MS2-reporters, NIH/3T3 cells were infected with lentivirus expressing tdMCP-GFP and GFP-expressing cells with low level of GFP expression were sorted by FACS. This stable population was infected with pInducer20-based reporter constructs carrying 24xMS2 binding sites, and stable lines were selected with geneticin (Thermo Fisher Scientific). Derived cell lines are: tdMCP-GFP_pIND20-b24bs/Net1 (MS2 reporter/Net1 UTR) and tdMCP-GFP_pIND20-b24bs/Rab13 (MS2 reporter/Rab13 UTR). Expression of the reporters was induced by addition of 1 μg/ml Doxycycline (Fisher Scientific) approximately 2–3 hr before imaging.

For translation inhibition, cells were treated with 100 μg/ml cycloheximide (Sigma Aldrich, Cat# 239763), 2 μg/ml harringtonine (LKT Labs, product ID H0169), 1 μM lactimidomycin (Fisher Scientific, cat # 50-629-10001), 50 μg/ml anisomycin (Sigma Aldrich, Cat# A5862), 100 μg/ml puromycin (Thermo Fisher Scientific, Cat# A1113803). For knockdown experiments, 40 pmoles of siRNAs were transfected into cells with Lipofectamine RNAiMAX (Thermo Fisher Scientific, cat# 13778–150) according to the manufacturer's instructions. Cells were assayed after three days. siRNAs used were: AllStars Negative control siRNA (cat# 1027281) and si-Rab13 #8 (cat# SI02662702; target sequence: 5'-ATGGTCTTTCTTGGTATTAAA-3') from Qiagen.

## Protrusion/cell body isolation and RNA analysis

Protrusions and cell bodies were isolated from serum-starved cells plated for 2 hr on Transwell inserts equipped with 3.0 μm porous polycarbonate membrane (Corning) as previously described (*Wang et al., 2017*). Briefly, 1.5 million cells were plated per 25 mm filter and 1 or three filters were used for cell body or protrusion isolation, respectively. LPA was added to the bottom chamber for 1 hr and the cells were fixed with 0.3% paraformaldehyde for 10 min. For isolating protrusions, cell bodies on the upper surface were manually removed by wiping with cotton swab and laboratory paper. The protrusions on the underside were then solubilized by immersing the filter in crosslink reversal buffer (100 mM Tris pH 6.8, 5 mM EDTA, 10 mM DTT and 1% SDS) and gentle scraping. Cell bodies were similarly isolated after manually removing protrusions from the underside of the membrane. The extracts were incubated at 70°C for 45 min and used for RNA isolation using Trizol LS (Thermo Fisher Scientific).

For nanoString analysis, RNA samples were analyzed using a custom-made codeset and the nCounter analysis system according to the manufacturer's instructions.

## Polysome gradient analysis

Cells were plated the day before so that they were actively growing on the day of the assay. Cells were treated with 50 µg/ml cycloheximide for 30 min at 37°C and cytoplasmic extract was collected essentially as described in *Bor et al. (2006)*. 10–45% sucrose gradients were prepared using Bio-Comp gradient master (BioComp Instruments, Canada) according to the manufacturer's protocol and centrifuged at 37,000 rpm for 2 hr at 4°C in a SW41Ti rotor. After centrifugation, gradients were fractionated, and UV absorbance profiles were recorded using the BioComp piston gradient fractionator (BioComp Instruments, Canada). Based on the recorded UV profiles, fractions were pooled into the four sections described in the text. 20 ng of in vitro transcribed GFP RNA was added to each pooled fraction to correct for RNA recovery. RNA was isolated using Trizol LS (Thermo Fisher Scientific). RNA levels were assessed by nanoString analysis (nanostring Technologies, Seattle, WA) using a custom-made codeset and the nCounter analysis system according to the manufacturer's instructions.

## Fluorescence in situ hybridization (FISH)

FISH was performed with ViewRNA ISH Cell Assay kit (Thermo Fisher Scientific) according to the manufacturer's instructions. The following Affymetrix probe sets were used: *Net1* cat# VB1-3034209, *Cyb5r3* Cat# VB1-18647, *Cenpb* cat# VB1-18648, mouse *Rab13* cat# VB1-14374, human *Rab13* cat# VA1-12225, *Pkp4* Cat# VB4-600264, *Kif18b* cat# VA6-3170686, *Ddr2* cat# VB1-14375, *Arpc3* cat# VB1-14507, *P4hb* cat# VB6-15898. To detect polyA RNAs, LNA modified oligodT probes (30 nucleotides) labeled with ATTO 655 were added during hybridization, pre-amplification, amplification and last hybridization steps of ViewRNA ISH Cell Assay. Cell mask stain (Thermo Fisher Scientific) was used to identify the cell outlines. Samples were mounted with ProLong Gold antifade reagent (Thermo scientific).

## Puromycylation and proximity ligation detection (puro-PLA)

Cells plated on Fibronectin (Sigma, Cat# F1141)- or Collagen IV (Sigma, Cat# C5533)-coated coverslips were pre-treated (or not) with the indicated translation inhibitors for 15 min and incubated for 5 min with 100 µg/ml puromycin at 37C. Subsequent steps were based on the protocol by *tom Dieck et al. (2015)* with some modifications. Specifically, after puromycin incubation cell were quickly placed on ice, rinsed with ice-cold PBS, incubated for 2 min on ice with permeabilization buffer (50 mM Tris-Cl, pH 7.5, 5 mM MgCl$_2$, 25 mM KCl, 100 µg/ml cycloheximide, 0.15 mg/ml digitonin, 0.5 U/µl RNasin, and Halt protease inhibitor cocktail), washed twice with ice-cold permeabilization buffer and fixed with 4% formaldehyde in PBS for 15 min at room temperature. Cells were subsequently blocked in blocking buffer (5% goat serum in PBS) for 1 hr at 37C and incubated with a pair of primary antibodies diluted in blocking buffer (in a humidified chamber for 1.5 hr at room temperature). Antibodies used were anti-Rab13 (1:200; Novus Biologicals, cat# NBP1-85799) and anti-puromycin 3RH11 (1:2,000, Kerafast, cat# EQ0001). After washing, PLA probes were applied in 1:10 dilution using the diluent buffer provided in the Duolink In Situ Red kit (Sigma Aldrich, cat# DUO92101). Incubations and subsequent ligation and amplification steps were performed according to the manufacturer's instructions. After the final washes, cells were post-fixed for 10 min at room temperature with 4% formaldehyde in PBS, stained with Alexa Fluor-488 phalloidin (Thermo Fisher, cat# A12379) in blocking buffer for 30 min and mounted using Duolink PLA Mounting medium with DAPI.

## Imaging and image analysis

FISH and puro-PLA images were obtained using a Leica SP8 confocal microscope, equipped with a HC PL APO 63x oil CS2 objective. Z-stacks through the cell volume were obtained and maximum intensity projections were used for subsequent analysis. Surface plot profiles, number and intensity of detected particles, were derived using the 'Surface plot' and 'Analyze particles' functions of ImageJ software (version 2.0.0-rc-69/1.52 n). Calculation of PDI index was performed using a custom Matlab script. The code is described and is available in *Stueland et al. (2019)*.

For live imaging of Lifeact-expressing cells, cells were plated on LabTek chambered coverglass, in phenol red-free media and were imaged on a Leica SP8 confocal microscope equipped with HC PL APO 63x oil CS2 objective, at constant 37C temperature and 5% CO2 (for NIH/3T3 and human dermal fibroblasts) or atmospheric air (for MDA-MB-231 cells). The 488 nm laser line was used for illumination, z-stacks through the cell volume were acquired over time and maximal intensity projections were produced.

For live imaging of RNA reporters, cells were plated on LabTek chambered coverglass, in phenol red-free media, and were imaged on a Zeiss LSM 780 confocal microscope, equipped with a Plan-Apochromat 63x/1.40 Oil M27 objective, at constant 37C temperature and 5% $CO_2$. 488 nm and 561 nm laser lines were used for illumination and single z-plane images were acquired over time. Two channel imaging was performed with sequential line scanning. Acquisition speed was approximately 500 ms/frame. No detectable photobleaching was observed under these conditions during a 20–40 s acquisition period. Diffraction-limited spots were identified using a custom IDL software provided by Dr. Daniel Larson (*Coulon et al., 2014*). The integrated fluorescence intensity of each spot was calculated using a Gaussian mask fit after local background subtraction (*Coulon et al., 2014*). Distance of spots from a user-defined protrusive edge were derived using a custom Matlab script, available upon request. We note that because of low signal to noise of the RNA (mCherry) channel, we were not confident of our ability to identify fast moving particles. Thus, we limited our analysis to more stationary particles defined as those particles that persist over >6 imaging frames.

## Acknowledgements

This work was supported by the Intramural Research Program of the Center for Cancer Research, NCI, National Institutes of Health (SM).

## Additional information

### Funding

| Funder | Grant reference number | Author |
| --- | --- | --- |
| National Cancer Institute | Intramural Research Program of the Center for Cancer Research | Stavroula Mili |

The funders had no role in study design, data collection and interpretation, or the decision to submit the work for publication.

### Author contributions

Konstadinos Moissoglu, Kyota Yasuda, Designed and performed experiments, analyzed data and edited the manuscript; Tianhong Wang, Formal analysis, Investigation, Methodology, Performed experiments and analyzed data; George Chrisafis, Performed experiments analyzed data and edited the manuscript; Stavroula Mili, Supervision, Funding acquisition, Designed, performed experiments, analyzed data, supervised the study, wrote and edited the manuscript

### Author ORCIDs

Konstadinos Moissoglu (iD) https://orcid.org/0000-0001-7211-4320
Kyota Yasuda (iD) https://orcid.org/0000-0002-8352-631X
Stavroula Mili (iD) https://orcid.org/0000-0002-9161-8660

### Decision letter and Author response

Decision letter https://doi.org/10.7554/eLife.44752.038
Author response https://doi.org/10.7554/eLife.44752.039

## Additional files

### Supplementary files
• Transparent reporting form
DOI: https://doi.org/10.7554/eLife.44752.036

### Data availability
All data generated or analysed during this study are included in the manuscript and supporting files.

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
