## [Decision Letter]

Thank you for submitting your article "Translational regulation of protrusion-localized RNAs involves silencing after transport" for consideration by *eLife*. Your article has been reviewed by two peer reviewers, and the evaluation has been overseen by a Reviewing Editor and James Manley as the Senior Editor.

The reviewers have discussed the reviews with one another and the Reviewing Editor has drafted this decision to help you prepare a revised submission.

The key to the work is the use of the SunTag system to identify localized mRNAs that have been or are being translated. The authors find that mRNAs appear to be translated as they are transported through the cytoplasm on their way to being localized, but then cluster and become translationally silenced. This goes against a prevailing view that mRNAs bound to the periphery are translationally repressed until they are localized. The reviewers find the work interesting but raise several concerns.

The following significant concerns must be addressed in a revised manuscript, should you wish to resubmit:

The reviewers agreed that the authors make generalizations that are not supported by the data. For instance, that they see mRNAs being translated while moving, these appear to be rare cases, and do not apply to all the mRNAs without necessary statistical analysis. The non-moving mRNAs may be more likely to translate, but there is a lack of quantification to evaluate this possibility. (They should cite the Wu et al. Science paper which also noted translating mRNAs in neurons-in fact all four of the single mRNA translation papers should be cited, not just Yan et al). Second, they do not consider alternative interpretations, for example that the translation is stalled until the mRNAs reach their destination, and the proteins are released there. More quantitation is needed before conclusions can be drawn as to polysome loading, for instance using qPCR. There is a noticeable lack of free protein, this seems to be a conundrum, along with the observation that translation is silenced exactly where it is needed. Discussion of the significance of this counter-intuitive observation is lacking. Finally, no consideration is given to the nature of the cells analyzed, and whether these results are an artifact of the cells or reporter chosen, The cells used, 3T3, are not good examples of cells with protrusive activity, or of polarized cells in general where localization is essential for motility (and it is known that immortalized cells localize mRNA poorly), and the reporter is not an endogenous gene with an endogenous promoter. They should test their system on primary cells to see if their observations hold there. So while the observations are potentially interesting they must be tempered by these specific constraints. The Discussion either needs to be softened, or the authors must provide additional data to address the generality of their observations. Finally, what conclusions the authors do want to draw on their specific system needs to be backed up by quantitation.

*Reviewer #1:*

This manuscript by Yasuda et al. have employed single molecule imaging of mRNA reporters to uncover a novel and very interesting mechanism for translation regulation of protrusion-localized mRNAs in cultured fibroblasts. In contrast to other reports that indicate transported mRNAs are translationally repressed during transport and unsilenced at the destination, here the data show that translational repression occurs after transport and involves the clustering of multiple mRNAs into heterogenous granules. The present experiments using fluorescent reporters is complemented by a biochemical approach to perturb mRNA localization which does not seem to alter translation. The strength of this paper is the use of SunTag reporters for live cell imaging of mRNA localization and translation, which does support a model for the formation of translationally repressed mRNPs within protrusions. While this finding does advance the state of knowledge for the field, there are concerns with some of data and resulting interpretations that suggest lack of translational repression during active mRNA transport. It does seem that new experiments could provide stronger evidence of mRNA silencing during active transport.

Specific comments:

Figure 1 shows that disruption of endogenous APC-dependent mRNA localization to protrusions using a competitive Pkp4 3'UTR reporter does not result in a shift of mRNA levels from polysome to mRNP fractions on sucrose gradients, in contrast to what was observed with puromycin treatment. The data need to be quantitated by real time PCR for several localized mRNAs. Specifically, the authors should report the actual values for% mRNA in each of the four fractions for several examples of localized mRNAs. While puromycin treatment resulted in marked mRNA shifts from heavy to light fractions, the competitive 3'UTR does appear to accumulate mRNA in an intermediate fraction #3. Another interpretation is that these are stalled ribosomes. This possibility has not been considered or discussed.

Figure 2 – the same comment as stated above also relates to use of pharmacologic perturbation of detyrosinated microtubules.

Figure 4 shows data obtained with the Sun-Tag reporter implying that the majority of single mRNAs have comparable translation efficiencies regardless of localization. The authors conclude that "RNAs targeted to protrusions are not silenced during transport." One major concern with this strong interpretation is that the quantitative analysis was not done on "directed" particles but rather on the majority of particles that were stationary or oscillatory. A qualitative example shown does suggest translation can occur during transport. Can an experiment be done to stimulate mRNA transport and increase the% of directed RNAs, for example, serum starvations followed by stimulation as done by Singer and colleagues for b-actin? Quantitative analysis of SunTag reporter dynamics can also be done following depolymerization of detyrosinated MTs. Without evidence on quantitation of mRNAs that are actively being transported, the authors will need to soften their conclusions. Such RNAs that are not moving and not localized may be translated, but this does not imply a similar fate for those that are actively being transported to the cell protrusion.

In the Discussion, the authors should not generalize their findings in fibroblasts to all polarized cells, but rather restrict their interpretation to cells that localize mRNAs as single mRNPs. In neurons, for example, some types of mRNA granules contain multiple mRNAs which do appear to be silenced during transport. These different models should be compared.

*Reviewer #2:*

Many localized mRNAs are translationally silenced during transport and become activated only upon reaching their cellular destination. In their manuscript, Yasuda et al. present evidence that a specific set of mRNAs, previously characterized as APC-dependent and associated with detyrosinated microtubules, is translationally active both during transport and at the cellular periphery. However, once at the periphery, a subset of APC-dependent RNAs coalesce into translationally inactive clusters which the authors speculate are needed for dynamic regulation during cell migration. The authors use translation reporters containing the SunTag peptide array and PP7 hairpins coupled with live cell imaging to reach the conclusion that the 3′ UTRs of several APC-dependent RNAs confer translation during transport and silencing in large clusters near the periphery. The clustering effect is also seen by FISH against endogenous APC-dependent RNAs. This paper provides an interesting example that runs counter to the dominant paradigm of translationally silent mRNA transport. However, some figures seem incomplete, for example lacking comparison to the control reporter. The discussion could also be made much stronger by further explaining the authors' hypothesis about the function of the peripheral RNA clusters. Are the clusters an intermediate in an RNA degradation pathway or are they RNA storage granules with translationally-competent RNAs that can be dispersed in a regulated fashion? These points are discussed in more detail below.

Major points:

1) Figure 5 presents one of the most interesting findings of the paper – that APC-dependent RNAs form clusters at protrusions. However, the presentation does not come across as rigorous and should be improved by: 1) quantification of cluster sizes and statement of number of cells analyzed and 2) comparison to the control reporter in the same figure. It seems that Figure S4 makes this argument in a more rigorous way. Perhaps part of Figure S4 could be moved to Figure 5 and part of Figure 5 could be moved to S4 (the polyA and the MS2-reporter panel could be moved to supplemental). It is additionally confusing that Figure S4 suggests that most RNAs are not in clusters but Figure 5 shows examples where almost no non-clustered RNAs can be seen. It seems like the images in Figure 5 must be the more exceptional examples. Therefore, some statement or quantification of the frequency of clusters from the live cell imaging is needed.

2) Similar to the previous point, Figure 6 should state how many cells were analyzed and include the control reporter as a comparison.

3) It would be interesting to see more analysis of polyA RNA clusters vs. APC-dependent RNA clusters. It is shown in one example for Pkp4 RNA in Figure 5E, but not quantified and not shown for other RNAs. I believe the authors' hypothesis is that the RNA clusters are translationally repressed, but still translationally competent (i.e. not degradation products). If the authors could show that most APC-dependent RNA clusters are polyA+, this finding would support that hypothesis.

4) Similar to point 4, the discussion does not explain the authors' hypothesis about the function of the RNA clusters clearly and why they are translationally silenced. There is only one sentence suggesting that they reflect 'a local regulatory mechanism used to silence APC-dependent RNAs'. It would be good to elaborate on this point and maybe offer a brief comparison to other types of translationally silent RNA granules.

---

## [Author Response]

The following significant concerns must be addressed in a revised manuscript, should you wish to resubmit:The reviewers agreed that the authors make generalizations that are not supported by the data. For instance, that they see mRNAs being translated while moving, these appear to be rare cases, and do not apply to all the mRNAs without necessary statistical analysis. The non-moving mRNAs may be more likely to translate, but there is a lack of quantification to evaluate this possibility. (They should cite the Wu et al. Science paper which also noted translating mRNAs in neurons-in fact all four of the single mRNA translation papers should be cited, not just Yan et al).

We completely agree with these comments and realized that some of the conclusions in the original manuscript were awkwardly stated. Our intent was not to conclude that RNAs are being translated *while* engaged in active transport, but rather that they can be translated similarly in both internal and peripheral locations. We have changed the text throughout the manuscript to clarify this. We have also specifically pointed out that the translation reporter imaging assay does not allow us to make accurate conclusions about the behavior of fast-moving molecules (subsection “RNAs targeted to protrusions are similarly translated in both internal and peripheral locations”) and have referenced the papers mentioned above. Furthermore, we have now used puro-PLA to detect in situ nascent Rab13 protein distribution (new Figures 7 and 8) (subsections “Endogenous Rab13 RNA is translated in both internal and peripheral locations” and “Peripheral Rab13 RNA is silenced at retracting protrusions”). This approach, apart from looking at the translation of an endogenous RNA, also circumvents the problem of fast-moving molecules by allowing us to look at translation in situ, in fixed samples. The results are fully consistent and support the conclusion that protrusion-localized RNAs can be translated in both peripheral and internal locations.

Second, they do not consider alternative interpretations, for example that the translation is stalled until the mRNAs reach their destination, and the proteins are released there.

To address this possibility, we have performed run-off experiments, by brief treatment (15min) with harringtonine or lactimidomycin, and now show that the detected translation signals largely reflect active translation. Specifically, the new Figure 3—figure supplement 1 (described in subsection “Single-molecule translation reporters of protrusion-localized RNAs”) shows that the signal detected by the translation imaging reporters is reduced almost to background levels upon harringtonine (or lactimidomycin), but not upon cycloheximide treatment, showing that it reflects active translation. Additionally, detection of Rab13-puro PLA signal is reduced to background level upon harringtonine treatment (compare harringtonine with anisomycin treatment) (Figure 7B and Figure 8C) showing that endogenous Rab13 is also mostly translated by actively scanning ribosomes (new Figures 7 and 8) (subsections “Endogenous Rab13 RNA is translated in both internal and peripheral locations” and “Peripheral Rab13 RNA is silenced at retracting protrusions”). We have also seen that RNAs fractionating in polysomal fractions of sucrose gradients are shifted towards the non-translating fractions of the gradient upon harringtonine treatment. We have seen this result for a few RNAs we have detected by RT-PCR in human breast cancer cells (as detailed below we see very similar mechanisms in all cell types tested). These results are shown in Author response image 1.

**Author response image 1. respfig1:** MDA-MB-231 cells were treated, or not, with harringtonine for 15min. Fractions of polysome gradients were isolated and analyzed by RT-ddPCR to detect two localized, APC-dependent RNAs (Rab13 and Net1) and a non-localized control RNA (GAPDH).

Unfortunately, due to technical issues we have been unable to perform this experiment in mouse fibroblasts with the full panel of RNAs analyzed in Figures 1 and 2. However, given the effect of harringtonine seen in all other assays, we believe that even without this experiment we provide strong support for the conclusion that stalled ribosomes are not a significant part of the described mechanism.

More quantitation is needed before conclusions can be drawn as to polysome loading, for instance using qPCR.

We believe that we might have not clearly conveyed that the heatmaps shown in Figures 1 and 2 are indeed averages of independent experiments. The associated p-values were meant to reflect whether the underlying distributions were different or not. We have now supplied, as source data, files with the exact values of all replicates used for the generation of the heatmaps and statistics in those figures.

With regards to the use of qPCR, we have opted to quantify RNA abundance using nanoString analysis instead of RT-qPCR because it allows for direct RNA counting and thus avoids biases introduced by reverse transcription and amplification. This is now stated in subsection “Disrupting the localization of APC-dependent RNAs at protrusions does not affect their translation”. Indeed, when we detect the spike RNA (which is added in equal amounts in the collected sucrose gradient fractions) we find that it is much more consistently and uniformly detected by nanoString analysis. RT-PCR detection shows much larger variations in the detected spike (even though the same amount was added). This underscores the variability and biases that can result from reverse transcription and amplification, which we think are exaggerated in samples with very different RNA complexities, such as those isolated from sucrose gradients.

There is a noticeable lack of free protein, this seems to be a conundrum, along with the observation that translation is silenced exactly where it is needed. Discussion of the significance of this counter-intuitive observation is lacking.

With regards to the lack of free protein, with think that this is probably a perception caused by the contrast of some images, which makes the free protein not so evident. However, this is not a real issue with the assay. Even though we specifically induce the expression of the reporters for only a short period of time (2-4hrs), precisely to avoid accumulation of both the RNA and protein to high levels, we nevertheless always detect a significant amount of free protein, seen as fast-moving GFP particles that are brighter compared to the diffuse background of the single chain GFP-labeled antibody. We think this is evident, for example, in Figures 3, figure 3—figure supplement 1 and Figure 4.

With regards to the conundrum of silencing translation at the periphery, we had tried to argue before that this reflects regulation of translation during dynamic processes, however we fully understand that more and better evidence was needed. Therefore, we have now provided substantial new evidence that we believe convincingly shows that silencing occurs at retracting protrusions. Thus, the resolution of the conundrum that we propose is that RNAs are translated in extending protrusions/lamellipodia and are silenced in retracting protrusions. These data are based on the use of puro-PLA to detect nascent Rab13 protein in situ and are detailed in the new Figures 7 and 8 (subsections “Endogenous Rab13 RNA is translated in both internal and peripheral locations” and “Peripheral Rab13 RNA is silenced at retracting protrusions”).

Finally, no consideration is given to the nature of the cells analyzed, and whether these results are an artifact of the cells or reporter chosen, The cells used, 3T3, are not good examples of cells with protrusive activity, or of polarized cells in general where localization is essential for motility (and it is known that immortalized cells localize mRNA poorly)…

We have now provided data in primary human dermal fibroblasts as well as in breast cancer cells (MDA-MB-231). We note that for the group of APC-dependent RNAs we are studying, we see robust localization in all cell types we have tested (primary, immortalized or transformed). We specifically show that in all cell types tested we see very similar behaviors in terms of the localization of the studied RNAs at protrusions; their translation in both internal and peripheral regions; their silencing at the periphery; and their coalescence in heterogeneous granules (new Figures 7-9 and subsections “Endogenous Rab13 RNA is translated in both internal and peripheral locations”; “Peripheral Rab13 RNA is silenced at retracting protrusions” and “Silenced Rab13 RNA at retracting protrusions can be found in heterogeneous clusters”). We therefore believe that we provide strong evidence that the reported mechanism is not an artifact of the cell line used but is generally exhibited by mesenchymal migrating cells.

3T3s are indeed different than the other two cell types used, in that they tend to form more contractile stable protrusions whose dynamics are much slower (we have provided videos to illustrate that point; compare Videos 11 and 12 to Video 13). As we briefly discuss (Discussion paragraph seven), we believe these slower dynamics might be the reason for the more pronounced formation of peripheral granules in 3T3 cells compared to the granules formed in the much more dynamic MDA-MB-231s. We believe that all cell types can provide valid information and that each one, through accentuating certain behaviors, can offer a valuable model to facilitate the study of different aspects of these structures.

… and the reporter is not an endogenous gene with an endogenous promoter.

The point of how much any reporter reflects endogenous regulation is of course very valid. Even if a reporter recapitulates certain behaviors of endogenous RNAs (like we see for our reporters in terms of their peripheral localization and clustering), there is no guarantee that other aspects are also accurately reflecting the endogenous regulation. We find however, that it would be very difficult to ever provide unequivocal evidence that an exogenous reporter accurately reflects endogenous regulation, unless the regulation of the endogenous RNAs is first known. Therefore, instead of testing the effect of adding further elements (5’UTR, promoter, introns etc.) to our translation reporters, or of engineering aptamers into the endogenous genes, we rather chose to directly assess in situ the translation of an endogenous APC-dependent RNA. For this we used the puro-PLA method (puromycylation followed by proximity ligation amplification), which allows visualization of nascent proteins and translation sites of specific endogenous transcripts. In the new Figures 7-8 we provide controls for the validation of the assay in different cell types and use it to corroborate the findings obtained with the translation imaging reporters, as well as to extend them, showing that silencing occurs at retracting protrusions.

We are aware that some of the conclusions we have reached with these experiments could be further extended by live imaging of translation reporters in the more dynamic MDA-MB-231 cells. While these are certainly very interesting experiments that we intend to pursue in the future, we hope that you appreciate that generation of such stable lines, appropriate for high quality imaging, are a lengthy endeavor. We believe that the current data fully support the conclusions of the manuscript and hope you will find that the information contained in it is warranting publication at this stage.

They should test their system on primary cells to see if their observations hold there.

As mentioned above we have provided data in both primary cells and transformed cancer cells that are fully consistent with our previous conclusions.

So while the observations are potentially interesting they must be tempered by these specific constraints. The Discussion either needs to be softened, or the authors must provide additional data to address the generality of their observations. Finally, what conclusions the authors do want to draw on their specific system needs to be backed up by quantitation.

We have added quantitations that we believe support all the claims made.

Given the new data added, we have modified sections of the Abstract, Results and Discussion to reflect the new information included in the revised version. Please see manuscript text with tracked changes for exact modifications.

Reviewer #1:

[…] The strength of this paper is the use of SunTag reporters for live cell imaging of mRNA localization and translation, which does support a model for the formation of translationally repressed mRNPs within protrusions. While this finding does advance the state of knowledge for the field, there are concerns with some of data and resulting interpretations that suggest lack of translational repression during active mRNA transport. It does seem that new experiments could provide stronger evidence of mRNA silencing during active transport.Specific comments:Figure 1 shows that disruption of endogenous APC-dependent mRNA localization to protrusions using a competitive Pkp4 3'UTR reporter does not result in a shift of mRNA levels from polysome to mRNP fractions on sucrose gradients, in contrast to what was observed with puromycin treatment. The data need to be quantitated by real time PCR for several localized mRNAs. Specifically, the authors should report the actual values for% mRNA in each of the four fractions for several examples of localized mRNAs. While puromycin treatment resulted in marked mRNA shifts from heavy to light fractions, the competitive 3'UTR does appear to accumulate mRNA in an intermediate fraction #3. Another interpretation is that these are stalled ribosomes. This possibility has not been considered or discussed.

We believe that we might have not clearly conveyed that the heatmaps shown in Figures 1 and 2 are indeed averages of independent experiments expressing the% mRNA in each fraction. The associated p-values were meant to reflect whether the underlying distributions were different or not. We have now supplied, as source data, files with the exact values of all replicates used for the generation of the heatmaps in those figures.

We have detailed in point 3 our rationale for using nanoString instead of RT-qPCR for quantitation of RNA levels.

We have performed ribosome run-off experiments with harringtonine to show that stalled ribosomes are not a significant part of the described mechanism. Please see response to point 2 for details.

Figure 2 – the same comment as stated above also relates to use of pharmacologic perturbation of detyrosinated microtubules.

The heatmaps in this figure are also showing the averages of independent experiments expressing the% mRNA in each fraction. We have now supplied the source data with the exact values of all replicates used for the generation of the heatmaps in this figure.

Figure 4 shows data obtained with the Sun-Tag reporter implying that the majority of single mRNAs have comparable translation efficiencies regardless of localization. The authors conclude that "RNAs targeted to protrusions are not silenced during transport." One major concern with this strong interpretation is that the quantitative analysis was not done on "directed" particles but rather on the majority of particles that were stationary or oscillatory. A qualitative example shown does suggest translation can occur during transport. Can an experiment be done to stimulate mRNA transport and increase the% of directed RNAs, for example, serum starvations followed by stimulation as done by Singer and colleagues for b-actin? Quantitative analysis of SunTag reporter dynamics can also be done following depolymerization of detyrosinated MTs. Without evidence on quantitation of mRNAs that are actively being transported, the authors will need to soften their conclusions. Such RNAs that are not moving and not localized may be translated, but this does not imply a similar fate for those that are actively being transported to the cell protrusion.

We agree with this point and acknowledge that our conclusions were awkwardly stated. Please see response to point 1 for details.

In the Discussion, the authors should not generalize their findings in fibroblasts to all polarized cells, but rather restrict their interpretation to cells that localize mRNAs as single mRNPs. In neurons, for example, some types of mRNA granules contain multiple mRNAs which do appear to be silenced during transport. These different models should be compared.

It would indeed be interesting to explore how these RNAs are regulated in neuronal cells. We have added in the Discussion a specific mention to that effect and a statement emphasizing that our findings are describing behaviors of mesenchymal migrating cells (Discussion penultimate paragraph).

Reviewer #2:

[…] The clustering effect is also seen by FISH against endogenous APC-dependent RNAs. This paper provides an interesting example that runs counter to the dominant paradigm of translationally silent mRNA transport. However, some figures seem incomplete, for example lacking comparison to the control reporter. The discussion could also be made much stronger by further explaining the authors' hypothesis about the function of the peripheral RNA clusters. Are the clusters an intermediate in an RNA degradation pathway or are they RNA storage granules with translationally-competent RNAs that can be dispersed in a regulated fashion? These points are discussed in more detail below.Major points:1) Figure 5 presents one of the most interesting findings of the paper – that APC-dependent RNAs form clusters at protrusions. However, the presentation does not come across as rigorous and should be improved by: 1) quantification of cluster sizes and statement of number of cells analyzed and 2) comparison to the control reporter in the same figure. It seems that Figure S4 makes this argument in a more rigorous way. Perhaps part of Figure S4 could be moved to Figure 5 and part of Figure 5 could be moved to S4 (the polyA and the MS2-reporter panel could be moved to supplemental). It is additionally confusing that Figure S4 suggests that most RNAs are not in clusters but Figure 5 shows examples where almost no non-clustered RNAs can be seen. It seems like the images in Figure 5 must be the more exceptional examples. Therefore, some statement or quantification of the frequency of clusters from the live cell imaging is needed.

The confusion about the prevalence of clusters probably stems from the fact that Figure 5 showed enlargements of protrusive regions only, while Figure S4 showed whole-cell images and the associated quantifications. We have now noted this more obviously in both the text, figure legends and the figures themselves

We have also reorganized the data of Figure 5 and Figure S4 (now named Figure 5—figure supplement 1) and added new panels in order to present quantitations and statistical comparisons in the main figure (Figure 5H, I) (subsection “APC-dependent RNAs associate with heterogeneous clusters at the tips of protrusions”). We have tried throughout to mention sample sizes used for analyses.

2) Similar to the previous point, Figure 6 should state how many cells were analyzed and include the control reporter as a comparison.

We have included in the legend the number of cells imaged. We had not included the control reporter because we don’t see any clusters. In any case, the point of this figure is not to provide a quantitative assessment of cluster formation between control and localized reporters, but to compare the translation status of RNAs within clusters to single RNAs within the same protrusions.

3) It would be interesting to see more analysis of polyA RNA clusters vs. APC-dependent RNA clusters. It is shown in one example for Pkp4 RNA in Figure 5E, but not quantified and not shown for other RNAs. I believe the authors' hypothesis is that the RNA clusters are translationally repressed, but still translationally competent (i.e. not degradation products). If the authors could show that most APC-dependent RNA clusters are polyA+, this finding would support that hypothesis.

We have added further quantitation regarding the coincidence of polyA signal with APC-dependent RNA clusters (Figure 5I) to support the use of polyA staining as an identifier of peripheral clusters in 3T3 cells. Indeed, the majority of APC-dependent RNA clusters are polyA+, and polyA staining can even reveal clusters that don’t contain the particular APC-dependent RNA being imaged.

While it is of interest, we cannot currently make any assessment regarding the fate of RNA within granules. As mentioned in point 6, we are aware that information relevant to these questions could be provided by live imaging of translation reporters in the more dynamic MDA-MB-231 cells. While these are certainly very interesting experiments that we intend to pursue in the future, we hope that you appreciate that they are too involved to pursue in the context of the current manuscript. For now, we have added a mention to these issues in the Discussion (paragraph seven).

4) Similar to point 4, the discussion does not explain the authors' hypothesis about the function of the RNA clusters clearly and why they are translationally silenced. There is only one sentence suggesting that they reflect 'a local regulatory mechanism used to silence APC-dependent RNAs'. It would be good to elaborate on this point and maybe offer a brief comparison to other types of translationally silent RNA granules.

This point is now addressed more deeply with the addition of new data showing that silencing occurs at retracting protrusions (new Figures 7-9, subsections “Endogenous Rab13 RNA is translated in both internal and peripheral locations” and “Peripheral Rab13 RNA is silenced at retracting protrusions”).